# Quantifying land carbon cycle feedbacks under negative $CO_2$ emissions

V. Rachel Chimuka[1], Claude-Michel Nzotungicimpaye[1,a] & Kirsten Zickfeld[1]

[1]Department of Geography, Simon Fraser University, Burnaby, BC, V5A 1S6, Canada

[a] Now at Department of Geography, Planning and Environment, University of Concordia, Montréal, QC, H3G 1M8, Canada

*Correspondence to*: V. Rachel Chimuka (rchimuka@sfu.ca)

**Abstract.** Land and ocean carbon sinks play a major role in regulating atmospheric $CO_2$ concentration and climate. However, their future efficiency depends on feedbacks in response to changes in atmospheric $CO_2$ concentration and climate, namely the concentration-carbon and climate-carbon feedbacks. Since carbon dioxide removal is a key mitigation measure in emission

scenarios consistent with global temperature goals in the Paris Agreement, understanding carbon cycle feedbacks under negative $CO_2$ emissions is essential. This study investigates land carbon cycle feedbacks under positive and negative $CO_2$ emissions using an Earth system model of intermediate complexity (EMIC) driven with an idealized scenario of symmetric atmospheric $CO_2$ concentration increase (ramp-up) and decrease (ramp-down), run in three modes. Our results show that the magnitudes of carbon cycle feedbacks are generally smaller in the atmospheric $CO_2$ ramp-down phase than in the ramp-up

phase, except for the ocean climate-carbon feedback, which is larger in the ramp-down phase. This is largely due to carbon cycle inertia: the carbon cycle response in the ramp-down phase is a combination of the committed response to the prior atmospheric $CO_2$ increase and the response to decreasing atmospheric $CO_2$. To isolate carbon cycle feedbacks under decreasing atmospheric $CO_2$ and quantify these feedbacks more accurately, we propose a novel approach that uses zero emissions simulations to quantify the committed carbon cycle response. We find that the magnitudes of the concentration-carbon and

climate-carbon feedbacks under decreasing atmospheric $CO_2$ are larger in our novel approach than in the standard approach. Accurately quantifying carbon cycle feedbacks in scenarios with negative emissions is essential for determining the effectiveness of carbon dioxide removal in drawing down atmospheric $CO_2$ and mitigating warming.

## 1 Introduction

Anthropogenic $CO_2$ emissions have increased substantially since the preindustrial era, increasing the risk of "severe, pervasive and irreversible impacts" to the Earth system (IPCC, 2022). In an effort to reduce greenhouse gas emissions, nations adopted the Paris Agreement, which stipulated that surface warming should be kept well below 2°C above preindustrial levels and encouraged efforts to further limit it to 1.5°C (UNFCCC, 2015). Carbon dioxide removal (CDR) is a key mitigation measure in emission scenarios that are consistent with these climate goals (Ciais et al., 2013; Fuss et al., 2014; Rogelj et al., 2018; Rogelj et al., 2019; IPCC, 2022).

The land and ocean carbon sinks play a major role in regulating atmospheric $CO_2$ concentration by absorbing approximately half of current anthropogenic $CO_2$ emissions (Friedlingstein et al., 2022). However, this rate of absorption is sensitive to changes in climate and atmospheric $CO_2$ concentration (Cox et al., 2000; Boer & Arora, 2010; Arora et al., 2013; Boer & Arora, 2013; Arora et al., 2020). As atmospheric $CO_2$ concentration increases, carbon sinks will take up more carbon through air-sea exchange and $CO_2$ fertilization, resulting in a negative concentration-carbon cycle feedback (Boer & Arora, 2010; Arora et al., 2013; Schwinger & Tjiputra, 2018). Conversely, changing climate, in response to the increasing $CO_2$ concentration, will decrease the ability of carbon sinks to take up carbon, resulting in a positive climate-carbon cycle feedback (Cox et al., 2000; Jones et al., 2003; Fung et al., 2005; Friedlingstein et al., 2006; Boer & Arora, 2010; Zickfeld et al., 2011; Boer & Arora, 2013; Friedlingstein et al., 2014; Schwinger & Tjiputra, 2018).

Since the dominant feedback controlling land and ocean carbon uptake is the negative concentration-carbon feedback, the land and ocean are currently carbon sinks (Arora et al., 2020). However, the implementation of negative emissions is expected to weaken or even reverse natural carbon sinks. If negative emissions are implemented but remain lower than positive emissions (net-positive emissions), the land and ocean carbon sinks continue to take up carbon, albeit at a lower rate (Tokarska & Zickfeld, 2015; Jones et al., 2016; Melnikova et al. 2021, Koven et al., 2022). On land, the rate of carbon uptake declines because ecosystem respiration increases more than gross primary productivity increases, whereas, in the ocean, the rate of uptake declines following the declining $CO_2$ emissions growth rate (Melnikova et al., 2021). Once the amount of $CO_2$ removed from the atmosphere exceeds the amount of $CO_2$ added to the atmosphere (net-negative emissions), the carbon sinks are expected to weaken further and may reverse (Cao & Caldeira, 2010; Tokarska & Zickfeld, 2015; Jones et al., 2016; Melnikova et al,, 2021; Canadell et al., 2022; Koven et al., 2022). Decreasing $CO_2$ levels will weaken the $CO_2$ fertilization effect, decreasing net primary productivity (NPP) more than soil respiration, resulting in a flux of carbon into the atmosphere (Cao & Caldeira, 2010; Tokarska & Zickfeld, 2015). Furthermore, the gradient in the partial pressure of $CO_2$ at the atmosphere-ocean interface will weaken and eventually reverse, resulting in the outgassing of $CO_2$ (Cao & Caldeira, 2010; Tokarska & Zickfeld, 2015). Carbon losses from the land and ocean following CDR are expected to significantly decrease the effectiveness of CDR in drawing down atmospheric $CO_2$ (Tokarska & Zickfeld, 2015; Jones et al., 2016; Zickfeld et al., 2021).

The behaviour of land carbon cycle feedbacks under positive and negative emissions is shown qualitatively in Figure 1. As the atmospheric $CO_2$ concentration increases under positive emissions, the land sequesters more carbon, reducing the atmospheric $CO_2$ concentration (Boer & Arora, 2010; Arora et al., 2013). However, under negative emissions, the declining atmospheric $CO_2$ concentration weakens and eventually reverses the land carbon sink, returning $CO_2$ to the atmosphere. The concentration-carbon feedback is negative because it promotes carbon sequestration under positive emissions and drives carbon loss under negative emissions. As the climate warms under positive emissions, the land loses carbon to the atmosphere, increasing the atmospheric $CO_2$ and causing further warming (Cox et al., 2000; Jones et al., 2003; Fung et al., 2005; Friedlingstein et al., 2006; Boer & Arora, 2010; Zickfeld et al., 2011; Boer & Arora, 2013; Friedlingstein et al., 2014). With cooling, the land carbon source weakens and eventually turns into a carbon sink, sequestering carbon and further cooling the climate under negative emissions. This positive climate-carbon feedback acts to amplify warming under positive emissions and enhance cooling under negative emissions.

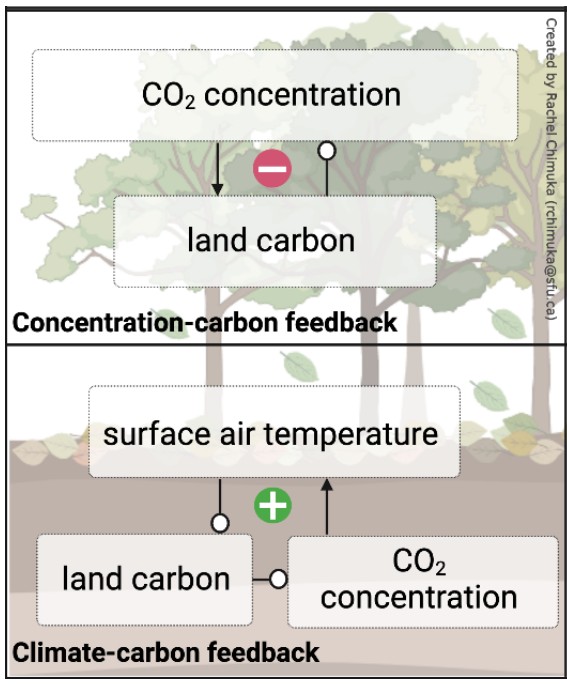

**Figure 1: Carbon cycle feedback schematic illustrating the behaviour of the negative concentration-carbon feedback (top box) and positive climate-carbon feedback (bottom box). Each feedback loop starts with an increase (under positive emissions) or decrease (under negative emissions) in atmospheric $CO_2$ concentration or surface air temperature. Arrows indicate a positive coupling (change in the same direction) between components and lines with empty circles indicate a negative coupling (change in the opposite direction) between components.**

The goal of this study is to quantify land carbon cycle feedbacks under negative emissions. We address two research questions:

(1) How does the magnitude of carbon cycle feedbacks under negative emissions compare to that under positive emissions?

(2) Is the approach currently used to quantify carbon cycle feedbacks under positive emissions adequate to quantify feedbacks

under negative emissions? If not, how can this approach be improved upon? This study investigates carbon cycle feedbacks under positive and negative emissions in an Earth system model of intermediate complexity (EMIC) driven with an idealized scenario with a 1% per year increase and decrease in atmospheric $CO_2$ concentration. Our study adds to the small but growing body of research on carbon cycle feedbacks under negative emissions (Schwinger & Tjiputra, 2018; Melnikova et al., 2021) by exploring the behaviour of these feedbacks, with a focus on land processes. We propose a novel approach for quantifying carbon cycle feedbacks under negative emissions and provide insight into the role of these feedbacks in determining the effectiveness of carbon dioxide removal in reducing $CO_2$ levels.

## 2 Methodology

### 2.1 Model Description

The University of Victoria Earth System Climate Model (UVic ESCM, version 2.10) (**figure 2**) is a model of intermediate complexity with a horizontal grid resolution of 1.8° (meridional) x 3.6° (zonal) (Weaver et al., 2001; Mengis et al., 2020). The model consists of a simplified atmospheric model, a 3D ocean general circulation model, including ocean inorganic and organic carbon cycle models, coupled to a dynamic-thermodynamic sea ice model, and a land surface model coupled to a vegetation model (including permafrost) (Mengis et al., 2020). The atmosphere is a 2D energy-moisture balance model with dynamical wind feedbacks. Atmospheric heat and freshwater are transported through diffusion and advection (Weaver et al., 2001), based on wind velocities prescribed from monthly climatological wind fields from NCAR/NCEP reanalysis data (Eby et al., 2013). The 19-layer 3D ocean general circulation model is based on the Geophysical Fluid Dynamics Laboratory (GFDL) Modular Ocean Model Version 2 (MOM2) (Pacanowski, 1995). The coupled dynamic-thermodynamic sea ice model simulates sea ice dynamics through elastic, viscous and plastic deformation and flow mechanisms (Weaver et al., 2001). Ocean carbon is represented by an inorganic ocean carbon model following the Ocean Carbon Model Intercomparison Protocol (OCMIP), and a NPZD (nutrient, phytoplankton, zooplankton, detritus) model of ocean biology simulating carbon uptake by the biological pump, accounting for phytoplankton light and iron limitations (Keller er al., 2012). The land surface model, based on the Hadley Centre Met Office Surface Exchange Scheme (MOSES), simulates the terrestrial carbon cycle and is coupled to the Top-Down Representation of Interactive Foliage and Flora including Dynamics (TRIFFID) model which simulates vegetation and soil carbon (Meissner et al., 2003). This model version also includes a permafrost carbon model in the soil module that simulates permafrost carbon through a diffusion-based scheme (MacDougall & Knutti, 2016).

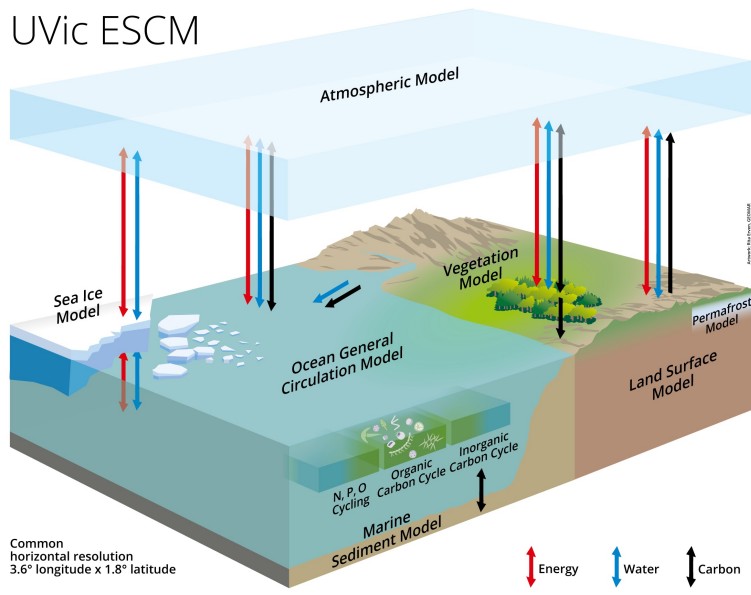

**Figure 2: University of Victoria Earth System Climate Model (UVic ESCM) schematic. Energy, water and carbon exchanges between model components are represented by arrows. Figure reproduced with permission from Mengis et al. (2020).**

### 2.2 Model Simulations

115   We performed a preindustrial spin-up simulation to equilibrate the model with the preindustrial $CO_2$ concentration (~285ppm). All other greenhouse gas concentrations, surface land conditions and orbital parameters were held at 1850 levels according to the Coupled Model Intercomparison Project Phase 6 (CMIP6) experimental design protocol (Eyring et al., 2016). The solar forcing was set to the 1850 – 1873 mean and the volcanic forcing was held at its average over 1850 – 2014, also consistent with CMIP6 protocol (Eyring et al., 2016).

120

To explore how the magnitude of carbon cycle feedbacks under positive emissions differs from that under negative emissions, we ran the "CDR-reversibility" simulation from the Carbon Dioxide Removal Model Intercomparison Project (CDRMIP) (Keller et al., 2018). Starting from a preindustrial equilibrium state, atmospheric $CO_2$ concentration was prescribed to increase at 1% per year until quadrupling, then decline back to preindustrial levels at the same rate. Achieving such a rapid decline in

125   $CO_2$ concentration would only be possible with substantial negative $CO_2$ emissions (Boucher et al., 2012). We refer to the section of the prescribed $CO_2$ concentration trajectory with increasing $CO_2$ concentration as the ramp-up phase and the section with decreasing $CO_2$ concentration as the ramp-down phase.

We also ran a zero emissions simulation ("Zeroemit") for use in our novel approach for quantifying the "committed" carbon

130   cycle response to increasing atmospheric $CO_2$ during the ramp-up phase. This simulation was initialized from the peak

atmospheric $CO_2$ concentration in the "CDR-reversibility" simulation and run in emissions-driven configuration. Emissions were set to zero at the start of the simulation, then $CO_2$ was allowed to evolve for 500 years.

The "CDR-reversibility" and "Zeroemit" simulations were run in three modes, following the C4MIP protocol for the quantification of carbon cycle feedbacks (Friedlingstein et al., 2006; Arora et al., 2013; Jones et al., 2016; Arora et al., 2020):

1. Fully coupled mode (FULL): the entire Earth system responds to the specified change in atmospheric $CO_2$ concentration or $CO_2$ emissions. In this mode, the land and ocean carbon sinks are subject to changing atmospheric $CO_2$ concentration and temperature.

2. Biogeochemically coupled mode (BGC): the land and ocean carbon sinks are subject to changing atmospheric $CO_2$ concentration but not changing temperature. This is achieved by prescribing a specified time-invariant $CO_2$ concentration to the radiation module (preindustrial $CO_2$ concentration for the "CDR-reversibility" simulation and quadruple the preindustrial $CO_2$ concentration for the "Zeroemit" simulation), while the land and ocean carbon cycle modules see an evolving atmospheric $CO_2$ concentration.

3. Radiatively coupled mode (RAD): the land and ocean carbon sinks are subject to changes in temperature but no change in atmospheric $CO_2$ concentration. The land and ocean carbon cycle modules see a specified time invariant $CO_2$ concentration (preindustrial $CO_2$ concentration in the "CDR-reversibility" simulation and quadruple the preindustrial $CO_2$ concentration in the "Zeroemit" simulation), while the radiation module sees changing atmospheric $CO_2$ concentration.

In both the "CDR-reversibility" and "Zeroemit" simulations, non-$CO_2$ forcings are held fixed at their preindustrial values.

**2.3 Approaches to Carbon Cycle Feedback Quantification**

In the first approach (referred to as the "standard" approach), we use the "CDR-reversibility" simulation to quantify carbon cycle feedbacks under increasing and decreasing atmospheric $CO_2$ concentration. Although this simulation is highly idealized, the ramp-up phase is standardly used to quantify carbon cycle feedbacks under positive emissions, and therefore, allows easier comparison of these results to other literature. The ramp-up phase represents the response to increasing atmospheric $CO_2$ alone. However, the ramp-down phase represents the response to both the prior increasing $CO_2$ and decreasing $CO_2$ because the latter is prescribed when the system is still in a transient (that is, time-evolving) state, responding to the prior atmospheric $CO_2$ increase (Zickfeld et al., 2016; Keller et al., 2018). As a result, carbon cycle feedbacks quantified from the ramp-down phase do not represent the response to decreasing atmospheric $CO_2$ alone.

Our second and novel approach, therefore, aims to improve the quantification of carbon cycle feedbacks under decreasing $CO_2$ by isolating the carbon cycle response to decreasing $CO_2$ alone. We use an experimental design utilizing both the "CDR-reversibility" and "Zeroemit" simulations. Since the "Zeroemit" simulation quantifies the "committed" or lagged response to

the prior positive emissions, the first 140 years of this simulation was subtracted from the ramp-down phase of the "CDR-
reversibility" simulation to isolate the response to decreasing $CO_2$ alone. A similar approach was used in Zickfeld et al. (2016)
to quantify the temperature response to decreasing atmospheric $CO_2$. The main assumption made here is that of linearity, that
is, we assume that the committed carbon cycle response to the prior $CO_2$ increase and the carbon cycle response to $CO_2$
decrease combine linearly to the total carbon cycle response in the ramp-down phase. From our approach – referred to as the
"inertia corrected" approach – we quantify carbon cycle feedbacks and compare them to those from the first approach.

**2.4 Carbon Cycle Feedback Metrics**

We use integrated flux-based feedback parameters (Friedlingstein et al., 2006) to quantify carbon cycle feedbacks in both
approaches. In this framework, changes in land and ocean carbon are expressed as the sum of two terms: a term representing
the change in land (ocean) carbon in response to changes in atmospheric $CO_2$, and a term representing the change in land
(ocean) carbon in response to changes in surface air temperature:


$$\Delta C_X = \beta_X \Delta C_A + \gamma_X \Delta T \qquad [1]$$

with the subscript $X$ representing land or ocean. The concentration-carbon feedback parameter $\beta$ quantifies the carbon cycle
response to changes in $CO_2$ concentration in units of PgC ppm$^{-1}$, whereas the climate-carbon feedback parameter $\gamma$ quantifies
the carbon cycle response to changes in climate in units of PgC $°C^{-1}$.

The change in land (ocean) carbon due to changing atmospheric $CO_2$ concentration is determined using the biogeochemically
coupled (BGC) simulation. In this simulation, the land and ocean only respond to changes in the $CO_2$ concentration, and
therefore, this simulation can be used to quantify the concentration-carbon feedback parameter $\beta$. Warming is still observed in
these simulations because the water use efficiency of vegetation increases at higher $CO_2$ concentrations and changes in albedo
due to shifts in vegetation structure and spatial distribution, result in a small warming effect (Cox et al., 2004, Boer & Arora,
2013; Arora et al., 2013). However, this warming is considered negligible in this framework (Friedlingstein et al., 2006).
Assuming that $\Delta T = 0$ in Eq. [1] ,the change in land (ocean) carbon due to changes in atmospheric $CO_2$ concentration is
expressed as:


$$\Delta C_X = \beta_X \Delta C_A \qquad [2]$$

Equation [2] can then be rearranged to solve for the concentration-carbon feedback parameter $\beta$ as follows:

$$\beta_X = \frac{\Delta C_X}{\Delta C_A} \qquad [3]$$


The change in land (ocean) carbon due to climate change is determined using the radiatively coupled (RAD) simulation. In this simulation, the land and ocean only respond to changes in climate, and therefore, this simulation can be used to quantify the climate-carbon feedback parameter $\gamma$. The change in land (ocean) carbon due to climate change is expressed as:


$$\Delta C_X = \gamma_X \Delta T \qquad [4]$$

Equation [4] can then be rearranged to solve for the climate-carbon feedback parameter $\gamma$ as follows:

$$\gamma_X = \frac{\Delta C_X}{\Delta T} \qquad [5]$$


An alternative method for quantifying the change in land (ocean) carbon due to climate change uses the fully coupled and biogeochemically coupled simulations (Arora et al., 2013). Here, we refer to this method as the FULL-BGC method. Here, the change in land (ocean) carbon in the biogeochemically coupled simulation (BGC) is subtracted from that in the fully coupled

simulation (FULL) and expressed as the product of the climate-carbon feedback parameter, and the difference between the surface air temperature changes in the two simulations:

$$\Delta C_X = \Delta C_X^{FULL} - \Delta C_X^{BGC} = \gamma_X (\Delta T^{FULL} - \Delta T^{BGC}) \qquad [6]$$

Equation [6] can then be rearranged to solve for the climate-carbon feedback parameter $\gamma$ as follows:

$$\gamma_X = \frac{C_X^{FULL} - \Delta C_X^{BGC}}{\Delta T^{FULL} - \Delta T^{BGC}} \qquad [7]$$

The resulting feedback parameters differ from those quantified from the RAD mode (Eq. [5]) alone due to nonlinearities in

carbon cycle feedbacks (Zickfeld et al., 2011; Schwinger et al., 2014).

Feedback parameters under increasing atmospheric $CO_2$ (ramp-up phase) are computed at the peak atmospheric $CO_2$ concentration (quadruple the preindustrial level) using changes in carbon pools, atmospheric $CO_2$ concentration and surface air temperature computed relative to preindustrial levels. Feedback parameters under decreasing atmospheric $CO_2$ (ramp-down

phase) are computed at the return to preindustrial levels (end of ramp-down phase) using changes in carbon pools, atmospheric $CO_2$ concentration and surface air temperature computed relative to the time of peak atmospheric $CO_2$.

In the ramp-up phase, feedback parameters are positive for land or ocean carbon gain and negative for land or ocean carbon loss. Note that the signs we refer to here are not the signs of the feedback but rather the signs of the feedback parameters, which are generally opposite to the sign of the feedback because they are computed from the perspective of the land and ocean, whereas the sign of the feedback is determined from the perspective of the atmosphere. In the ramp-down phase, both atmospheric $CO_2$ concentration and surface air temperature decline relative to their values at the end of the ramp-up phase, resulting in a negative denominator (see Eq. [3], [5], [7]). Therefore, the sign convention is reversed: feedback parameters are negative for a gain in land or ocean carbon (positive numerator divided by negative denominator) and positive for a loss in land or ocean carbon (negative numerator divided by negative denominator).

### 2.4.1 Isolating Carbon Cycle Feedbacks under Negative Emissions

When a $CO_2$ decrease is prescribed from a transient state, the land and ocean carbon pools not only respond to this $CO_2$ decrease, but also to the prior $CO_2$ trajectory due to inertia in these systems (Zickfeld et al., 2016). The land (ocean) carbon cycle responses in the ramp-down phase can, therefore, be expressed as the sum of two terms: one term driven by the sensitivities of land (ocean) to the $CO_2$ and temperature decrease during the ramp-down phase ('SENS' for sensitivity) and an inertia term that represents the lagged response to past atmospheric $CO_2$ and climate changes ('LAG'):

$$\Delta C_X = \Delta C_X^{SENS} + \Delta C_X^{LAG} \qquad [8]$$

The carbon pool response to the $CO_2$ and temperature decrease can then be isolated as follows:

$$\Delta C_X^{SENS} = \Delta C_X - \Delta C_X^{LAG} \qquad [9]$$

$\Delta C_X^{SENS}$ is driven by the sensitivities to changes in atmospheric $CO_2$ (β) and temperature (γ) in the ramp-down phase and can be linearly decomposed in the same way as the land and ocean carbon response in the standard framework (Eq. [1]):

$$\Delta C_X^{SENS} = \Delta C_X - \Delta C_X^{LAG} = \beta_X(\Delta C_A) + \gamma_X(\Delta T) \qquad [10]$$

Here, $\Delta C_A$ and $\Delta T$ refer to the changes in atmospheric $CO_2$ and temperature in the ramp-down phase of the CDR-reversibility simulation relative to their values at the end of the ramp-up phase. This framework becomes identical to the standard framework (section 2.4) in cases where a change in atmospheric $CO_2$ is applied from a state of equilibrium, i.e., $\Delta C_X^{LAG} = 0$.

Eq. [10] can be rewritten for the biogeochemically ($\Delta T = 0$) and radiatively coupled simulations ($\Delta C_A = 0$) respectively:

$$\Delta C_X^{SENS} = \Delta C_X - \Delta C_X^{LAG} = \beta_X(\Delta C_A) \qquad [11]$$

$$\Delta C_X^{SENS} = \Delta C_X - \Delta C_X^{LAG} = \gamma_X(\Delta T) \qquad [12]$$

Rearranging the equations above allows for the calculation of the feedback parameters, which measure the sensitivity of the land and ocean carbon response to changes in $CO_2$ concentration and temperature in the ramp-down phase:

$$\beta_X = \frac{\Delta C_X - \Delta C_X{}^{LAG}}{\Delta C_A} \qquad [13]$$

$$\gamma_X = \frac{\Delta C_X - \Delta C_X{}^{LAG}}{\Delta T} \qquad [14]$$

The lagged responses of land and ocean carbon pools $\Delta C_X^{LAG}$ are calculated from the "Zeroemit" simulations run in the respective mode (biogeochemically coupled for the calculation of $\beta$ and radiatively coupled for the calculation of $\gamma$), and are then subtracted from the responses of the ramp-down phase of the CDR-reversibility simulations run in the same mode. The land (ocean) carbon changes, surface air temperature and $CO_2$ concentration changes are computed relative to the year of peak $CO_2$ concentration (year 140 in the CDR-reversibility simulation; year 1 in the zero emissions simulation).

## 3 Results

### 3.1 "CDR-reversibility" Carbon Cycle Feedback Analysis

Our results focus on the ramp-down phase of the "CDR-reversibility" simulation and compare the system response in this phase to that in the ramp-up phase. While the prescribed atmospheric $CO_2$ concentration for the "CDR-reversibility" simulations is the same, the temperature response differs by mode (**figure 3(a, b)**). In the FULL and RAD modes, surface air temperature increases approximately linearly with increasing atmospheric $CO_2$ concentration, continues to increase for approximately half a decade after atmospheric $CO_2$ concentration peaks, then decreases with decreasing $CO_2$ concentration. Surface air temperature declines more slowly in the ramp-down phase due to the thermal inertia of the ocean, and therefore, does not return to preindustrial levels by the end of the ramp-down phase. The temperature response in the FULL mode is consistent with earlier studies (Boucher et al., 2012; Zickfeld et al., 2016; MacDougall, 2019; Ziehn et al., 2020; Park & Kug, 2022). Surface air temperature in the BGC mode changes only marginally: surface air temperature increases slightly with increasing $CO_2$ concentration and decreases as the $CO_2$ concentration decreases. This temperature change is driven by

biophysical responses to changing atmospheric $CO_2$, in particular, changes in evaporative fluxes as plants adjust stomatal conductance based on atmospheric $CO_2$ levels. Biophysical effects are also responsible for the difference in warming between the FULL and RAD modes (Arora et al., 2020). The temperature response in the ramp-up phase of the FULL, BGC and RAD

modes is consistent with Arora et al. (2020) while the temperature response in the ramp-up and ramp-down phases of all three modes is consistent with Schwinger & Tjiputra (2018).

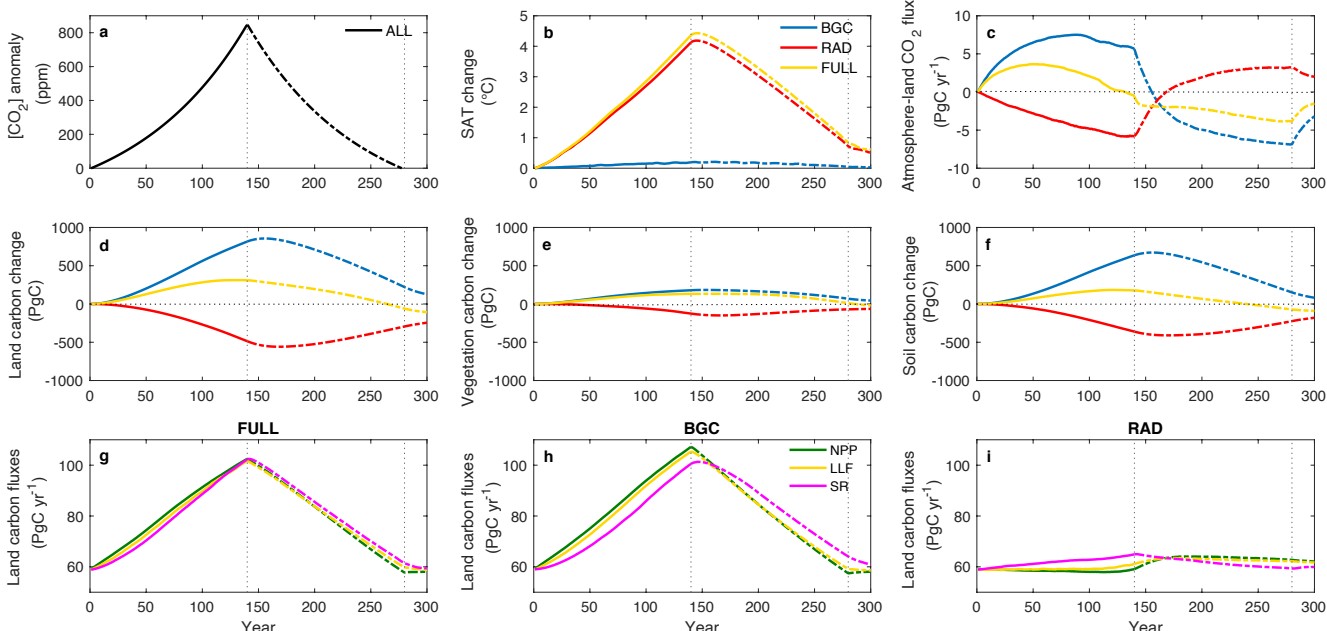

**Figure 3: a. Prescribed atmospheric $CO_2$ concentration anomaly b. surface air temperature change (SAT) c. atmosphere to land**
**$CO_2$ flux and d. land e. vegetation and f. soil carbon pool changes in the fully coupled (FULL), biogeochemically coupled (BGC) and radiatively coupled (RAD) "CDR-reversibility" simulations. Panels a, b, and d - f are calculated relative to 1850 (preindustrial). Carbon fluxes for the three modes are shown in the bottom panels (g, h, i). NPP = net primary productivity, LLF = leaf litter flux and SR = soil respiration. Solid lines represent the ramp-up phase and dot-dashed lines represent the ramp-down phase. The vertical dotted lines mark the beginning and end of the ramp-down phase.**

**3.1.1 Land Carbon Change in the FULL Mode**

**Figure 3(d)** shows land carbon pool changes as a function of time. In the FULL mode, land carbon increases, stabilizes, then begins to decrease 7 years before the peak atmospheric $CO_2$ concentration is reached. Similar carbon pool change patterns are observed for the soil carbon pool, which starts decreasing roughly 20 years before the peak in atmospheric $CO_2$ concentration, but vegetation carbon decreases 2 years after the peak atmospheric $CO_2$ concentration **(figure 3(e, f)).** Our results are

qualitatively consistent with Ziehn et al. (2020). However, they differ from other studies (MacDougall, 2019; Arora et al., 2020) wherein the land carbon pool remains a carbon sink in the ramp-up phase. MacDougall (2019) shows that the soil carbon

sink switches into a source later in the ramp-up phase than our results show. Furthermore, other studies (Boucher et al., 2012; Zickfeld et al., 2016) show that both vegetation and soil carbon sinks persist throughout the ramp-up phase.

Here, land carbon decreases throughout the ramp-down phase **(figure 3(d))** whereas, earlier studies show continued increase in the land carbon pool in the early ramp-down phase (Boucher et al., 2012; Zickfeld et al., 2016; Park & Kug, 2021). Changes in land carbon are governed by the balance between net primary productivity (NPP) and soil respiration. The increase in the land carbon pool is driven by the $CO_2$ fertilization effect: photosynthesis is enhanced under increasing $CO_2$ concentration, increasing NPP **(figure 3(g))** (Arora et al. 2013). Soil respiration also increases with warming **(figure 3(g))**. Initially, soil

respiration remains below NPP, but the rate of increase of NPP declines faster and soil respiration exceeds NPP towards the end of the ramp-up phase. This occurs due to the different response timescales of NPP and soil respiration: NPP depends on atmospheric $CO_2$ changes, whereas soil respiration depends on temperature change, which lags behind the change in $CO_2$ concentration (Cao & Caldeira, 2010). In the ramp-down phase, NPP decreases as the $CO_2$ fertilization effect weakens, whereas soil respiration continues to increase for a year before decreasing at a slower rate than NPP, driven by decreasing surface air

temperature and soil carbon.

### 3.1.2 Land Carbon Change in the BGC Mode

In the BGC mode, land carbon increases in the ramp-up phase, continues to increase until 16 years after the peak in $CO_2$ concentration, then decreases **(figure 3(d))**. A similar lag is observed for both vegetation and soil carbon pools, but the soil carbon sink persists for five years longer than the vegetation carbon sink **(figure 3(e, f))**. Land carbon increases in the ramp-

up phase due to the $CO_2$ fertilization effect, which increases NPP **(figure 3(h))** (Arora et al. 2013)**. In the UVic ESCM, soil respiration depends on soil temperature, moisture, and carbon content (Cox et al., 2001; Mengis et al., 2020). Since changes in surface air temperature in the BGC mode are small **(figure 3(b))**, changes in the first two factors are negligible and soil carbon content is the main driver of soil respiration changes. Soil respiration increases with increasing soil carbon, but NPP remains higher, resulting in an increase in the land carbon pool in the ramp-up phase **(figure 3(h))**. In the ramp-down phase,

NPP decreases as the $CO_2$ fertilization effect weakens, whereas soil respiration continues to increase before decreasing at a slower rate than NPP, following changes in soil carbon **(figure 3(h))**. NPP declines below soil respiration, and land carbon begins to decrease.

### 3.1.3 Land Carbon Change in the RAD Mode

Land carbon decreases in the ramp-up phase of the RAD mode, continues to decrease until roughly 30 years after the peak in

atmospheric $CO_2$ concentration, then switches into a carbon sink **(figure 3(d))**. Both vegetation and soil carbon pools exhibit a similar lag, but the vegetation carbon pool remains a carbon source for a decade longer than the soil carbon pool **(figure 3(e, f))**. Land carbon decreases in the ramp-up phase because NPP decreases as plant respiration rates increase **(see figure S1)**, whereas soil respiration increases with warming **(figure 3(i))** consistent with earlier literature (Arora et al., 2020). NPP later

increases due to vegetation shifts that occur on decadal to centennial timescales **(see figure S2)** but remains lower than soil respiration. In the ramp-down phase, NPP increases **(figure 3(i))** as gross primary productivity increases and plant respiration decreases with cooling, then later declines as gross primary productivity declines, because cooler temperatures negatively impact vegetation growth in the high latitudes **(see figures S1, S3).** Soil respiration decreases steadily with declining surface air temperature, and after a few decades, declines below NPP, and the land carbon pool begins to grow again.

### 3.1.4 Ocean Carbon Change in the FULL, BGC and RAD Modes

In the FULL mode, the ocean carbon pool grows at a steady rate, then begins to slowly lose carbon roughly three decades after the peak in atmospheric $CO_2$ concentration **(figure 4(a))**. In the ramp-up phase, the partial pressure of $CO_2$ in the atmosphere increases, strengthening the partial pressure gradient and driving an influx of $CO_2$ into the ocean **(figure 4(b)).** In the ramp-down phase, the gradient in partial pressure weakens and eventually reverses, and the ocean carbon sinks switches into a source. Earlier studies forced with the "CDR-reversibility" simulation also show ocean carbon uptake in the ramp-up phase (MacDougall, 2019; Arora et al., 2020) followed by delayed carbon loss in the ramp-down phase (Boucher et al., 2012; Zickfeld et al., 2016).

The ocean exhibits a delayed response in the ramp-down phase of the BGC and RAD modes consistent with Schwinger & Tjiputra (2018). In the BGC mode, ocean carbon increases in the ramp-up phase, continues to increase for approximately half a century after the peak atmospheric $CO_2$ concentration, then switches into a source of carbon **(figure 4(a)).** The partial pressure gradient of $CO_2$ strengthens in the ramp-up phase, driving $CO_2$ uptake, then weakens and reverses in the ramp-down phase, promoting carbon loss, but the magnitude of the flux is larger than in the FULL mode **(figure 4(b))**. In the RAD mode, ocean carbon decreases in the ramp-up phase, continues to decrease for over a century in the ramp-down phase, then switches into a weak carbon sink **(figure 4(a))**. The ocean outgasses in the ramp-up phase possibly due to climate effects on ocean circulation and the solubility pump (Cox et al., 2000; Fung et al., 2005; Friedlingstein et al., 2006; Zickfeld et al., 2011). In the ramp-down phase, the ocean remains a carbon source for over a century before switching into a weak carbon sink. Ocean carbon changes in the BGC and RAD modes are also driven by the concentration-carbon and climate-carbon feedbacks. An in-depth discussion of the mechanisms behind the ocean carbon response is beyond the scope of this paper.

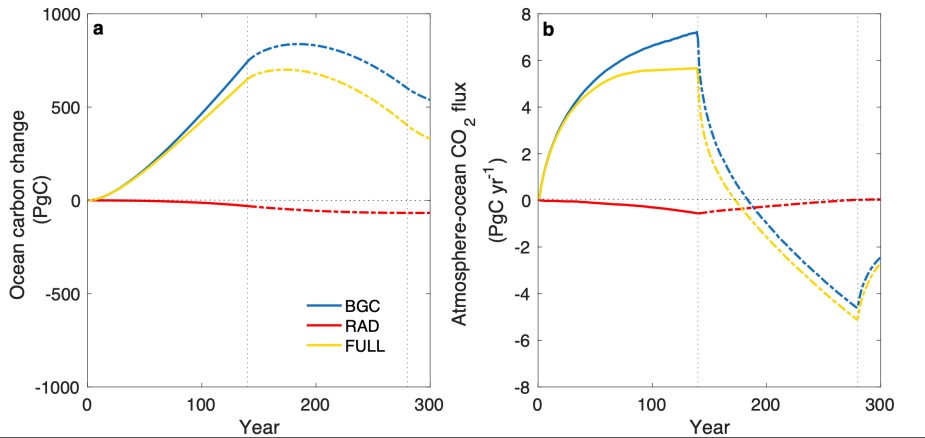

 **Figure 4: a. Ocean carbon change and b. atmosphere to ocean $CO_2$ flux in the fully coupled (FULL), biogeochemically coupled (BGC) and radiatively coupled (RAD) "CDR-reversibility" simulations. Ocean carbon change is calculated relative to 1850 (preindustrial). Solid lines represent the ramp-up phase and dot-dashed lines represent the ramp-down phase. The vertical dotted lines mark the beginning and end of the ramp-down phase.**

### 3.1.5 Sensitivity of Land and Ocean Carbon Pools

To assess the sensitivity of land and ocean carbon pools to changes in atmospheric $CO_2$ and temperature, we plot carbon changes in the BGC mode as a function of atmospheric $CO_2$ concentration **(figure 5)** and carbon changes in the RAD mode as a function of surface air temperature **(figure 6).** The trajectory of carbon change differs in the ramp-up and ramp-down phases of the BGC mode **(figure 5),** a behavior referred to as hysteresis. Hysteresis in the land carbon pool is primarily driven by the soil carbon pool, although the contribution from the vegetation carbon pool is also significant **(figure 5(a, c, d)).** The width of

the hysteresis – measured as the vertical distance between the ramp-up and ramp-down trajectories – initially increases, then decreases **(figure 5(a - d))**, except in the vegetation carbon pool where the width of the hysteresis increases throughout the ramp-down phase **(figure 5(c))**. The land and ocean carbon pools in the RAD mode also exhibit hysteresis **(figure 6).** The hysteresis in the land carbon pool is dominated by the soil carbon pool **(figure 5(d))**, and the width of the hysteresis appears to increase throughout the ramp-down phase for all carbon pools except the vegetation carbon, which shows nearly constant

hysteresis**.** The observed hysteresis in the land and ocean carbon pools in the BGC and RAD modes is likely largely due to climate system inertia: the carbon cycle response in the ramp-down phase is a combination of the response to both increasing and decreasing $CO_2$ concentrations.

Despite the restoration of preindustrial atmospheric $CO_2$ levels in the BGC mode, the land and ocean carbon pools do not

return to their preindustrial states. At the end of the ramp-down phase, the land carbon pool holds approximately 250 PgC more than at preindustrial, with 80 PgC remaining in vegetation and 170 PgC remaining in the soil **(figure 5(a, c, d)),** due to time lags associated with vegetation and soil carbon turnover. The ocean carbon pool holds much more carbon (615PgC) than at preindustrial **(figure 5(b)).** In the RAD mode, the land and ocean carbon lost in the ramp-up phase is not completely regained

in the ramp-down phase, though this response would be expected given the asymmetric surface air temperature response in
this mode. By the end of the RAD mode, the land carbon pool holds approximately 300 PgC less than at preindustrial, with
the vegetation carbon pool accounting for 70 PgC and the soil carbon pool accounting for the remaining 230PgC **(figure 6(a,
c, d))**. The ocean holds only 70PgC less than at preindustrial, but unlike the land carbon pool, a miniscule amount of ocean
carbon is regained in the ramp-down phase **(figure 6(b)).**

Previous studies have shown carbon cycle hysteresis in the FULL mode of the "CDR-reversibility" simulation (Boucher et al.,
2012; Zickfeld et al., 2016; Jeltsch-Thömmes et al., 2020; Park & Kug, 2022), consistent with our results (**see figure S4**).
However, in most of these studies, the vegetation and soil carbon pools do not return to their preindustrial states by the end of
the ramp-down phase (Boucher et al., 2012; Zickfeld et al., 2016; Park & Kug, 2022). Our results for the FULL mode of the
"CDR-reversibility" simulation show that the vegetation and soil carbon pools are very close to their preindustrial states by
the end of the ramp-down phase (**see figure S4**), consistent with Ziehn et al. (2020), who show a near-return to the preindustrial
state in the vegetation carbon pool.

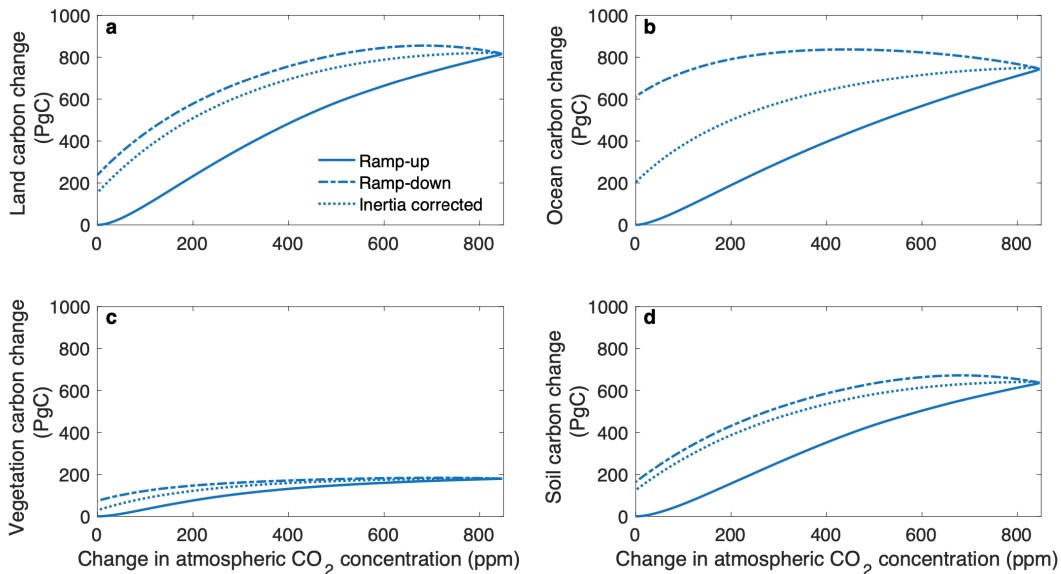

**Figure 5: a. Land b. ocean c. vegetation and d. soil carbon pool changes as a function of atmospheric CO₂ concentration, taken from
the biogeochemically coupled (BGC) "CDR-reversibility" simulation ramp-up and ramp-down phases, and "inertia corrected"**
**approach. All values are calculated relative to 1850 (preindustrial).**

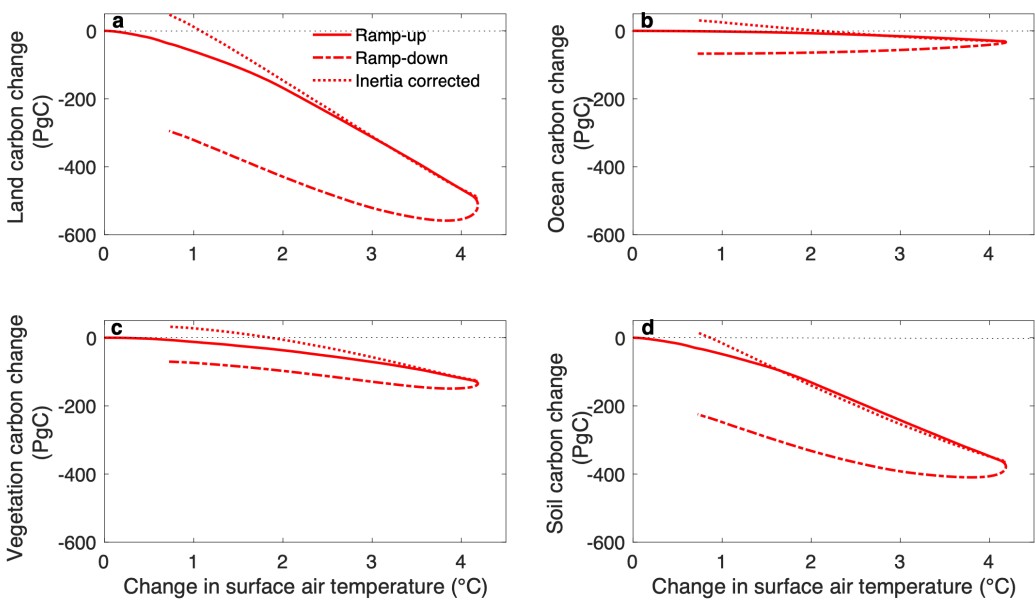

**Figure 6: a. Land b. ocean c. vegetation and d. soil carbon pool changes as a function of surface air temperature change, taken from the radiatively coupled (RAD) "CDR-reversibility" simulation ramp-up and ramp-down phases, and "inertia corrected" approach. All values are calculated relative to 1850 (preindustrial).**

### 3.1.6 Carbon Cycle Feedback Parameters quantified from "CDR-reversibility" simulations

**Table 1** shows the carbon cycle feedback parameters quantified using the Friedlingstein et al. (2006) carbon cycle feedback framework **(see Section 2.4)**. The concentration-carbon feedback parameter ($\beta$), which quantifies the concentration-carbon feedback, is computed as the change in land or ocean carbon per unit change in atmospheric $CO_2$ concentration in the BGC mode. The climate-carbon feedback parameter ($\gamma$) quantifies the climate-carbon feedback as the change in land or ocean carbon per unit change in surface air temperature in the RAD mode (referred to as the RAD approach). An alternative approach to quantifying the climate-carbon feedback involves taking the difference between the fully coupled and biogeochemically coupled simulations and computing the change in land or ocean carbon per unit change in surface air temperature from that difference (referred to here as the FULL-BGC approach).

In the "CDR-reversibility" simulation, the magnitudes of $\beta$ and $\gamma$ for both land and ocean are smaller in the ramp-down phase (under negative emissions) than in the ramp-up phase (under positive emissions), except the ocean climate-carbon feedback parameter, which is larger. (**Table 1**). Climate-carbon feedback parameters calculated using the FULL-BGC approach (shown in parentheses) are consistent in sign with those calculated using the RAD approach, but the magnitudes of these feedback parameters are larger (**see Figure S5** for hysteresis figures for this approach). Carbon cycle feedback parameters are smaller

in the ramp-down phase because the land and ocean carbon pools show a lagged response to changes in $CO_2$ concentration and climate in the early ramp-down phase. In the ocean, this lagged response to changes in climate is much greater, and carbon loss continues throughout the ramp-down phase (shown by the positive ocean climate-carbon feedback parameter). As a result, feedback parameters in the ramp-down phase are underestimated. Improving this quantification could be achieved by quantifying and removing this inertia.

| Simulations(s) used for calculation of feedback parameters | Positive Emissions (Ramp-up) | | | | Negative Emissions (Ramp-down) | | | |
|---|---|---|---|---|---|---|---|---|
| | $\beta_L$ | $\beta_O$ | $\gamma_L$ | $\gamma_O$ | $\beta_L$ | $\beta_O$ | $\gamma_L$ | $\gamma_O$ |
| | ($PgC\ ppm^{-1}$) | | ($PgC\ °C^{-1}$) | | ($PgC\ ppm^{-1}$) | | ($PgC\ °C^{-1}$) | |
| *"CDR-reversibility" simulation* taken at 4xCO₂ for positive emissions and at return to preindustrial for negative emissions | 0.96 | 0.88 | -117.8 (-121.5) | -7.36 (-22.7) | 0.68 | 0.16 | -56.4 (-67) | 10.8 (31.1) |
| *"Inertia corrected" approach* taken at 4xCO₂ for positive emissions and at return to preindustrial for negative emissions | 0.96 | 0.88 | -117.8 | -7.36 | 0.80 | 0.84 | -157.1 | -18.1 |

**Table 1: Carbon cycle feedback parameters under positive and negative emissions quantified at 4xCO₂ (quadruple the preindustrial CO₂ level) from the "CDR-reversibility" simulation and using the proposed "inertia corrected" approach. Feedback parameters for negative emissions are positive for land or ocean carbon loss and negative for land or ocean carbon gain, opposite to the sign convention for feedbacks under positive emissions. Values shown in parentheses were calculated using the FULL-BGC approach for quantifying climate-carbon feedbacks (see Eq. [7]). Feedback parameters quantified from the "CDR-reversibility" simulation can also be derived from Figures 5 and 6 respectively by taking the slope of the land or ocean response at the same time points at which they are computed.**

## 3.2 Isolating Carbon Cycle Feedbacks under Negative Emissions

### 3.2.1 "Zeroemit" Simulation: Quantifying Climate System Inertia

Zero emissions simulations quantify committed changes due to the prior $CO_2$ trajectory. Changes in atmospheric $CO_2$ concentration in zero emissions simulations are driven by the carbon sinks, which in turn are influenced by the $CO_2$ concentration and climate. Following cessation of emissions, the $CO_2$ concentration in the FULL mode declines steadily, mainly driven by ocean carbon uptake consistent with results from MacDougall et al. (2020) **(figure 7(a)).** The $CO_2$ concentration in the BGC mode declines more than in the FULL mode because both land and ocean remain carbon sinks. In the RAD mode, the $CO_2$ concentration increases as both land and ocean carbon decrease, releasing $CO_2$ into the atmosphere. Changes in atmospheric $CO_2$ concentration, together with changes in ocean heat uptake and surface albedo, drive changes in surface air temperature. In the FULL mode, the warming effect of declining ocean heat uptake dominates over the cooling

effect of declining $CO_2$ concentration resulting in continued warming (MacDougall et al., 2020) **(figure 7(b); figure S6).** The decline in $CO_2$ concentration is partly offset by permafrost carbon release from the soil **(figure 7(e))**. Surface air temperature in the RAD mode increases more than in the FULL mode because the $CO_2$ concentration increases, causing further warming. Surface air temperature remains relatively constant in the BGC mode. In the FULL mode, the land switches into a source of carbon after emissions cease, consistent with the behaviour of the UVic ESCM in the Zero Emissions Commitment Model

Intercomparison Project (ZECMIP) (MacDougall et al., 2020) **(figure 7(c)).** Vegetation carbon continues to increase **(figure 7(d))** whereas, soil carbon decreases **(figure 7(e))**. The ocean remains a carbon sink after cessation of emissions **(figure 7(f)).** In the BGC mode, the ocean remains a strong carbon sink after $CO_2$ emissions are set to zero, whereas land carbon initially increases, then decreases **(figure 7(c, f))**. Vegetation carbon increases throughout the zero emissions phase whereas, soil carbon initially increases, then slowly decreases **(figure 7(d, e))**. Both land and ocean carbon decrease in the RAD mode **(figure 7(c,**

**f))** with both vegetation and soil carbon pools driving this decrease **(figure 7(d, e)).**

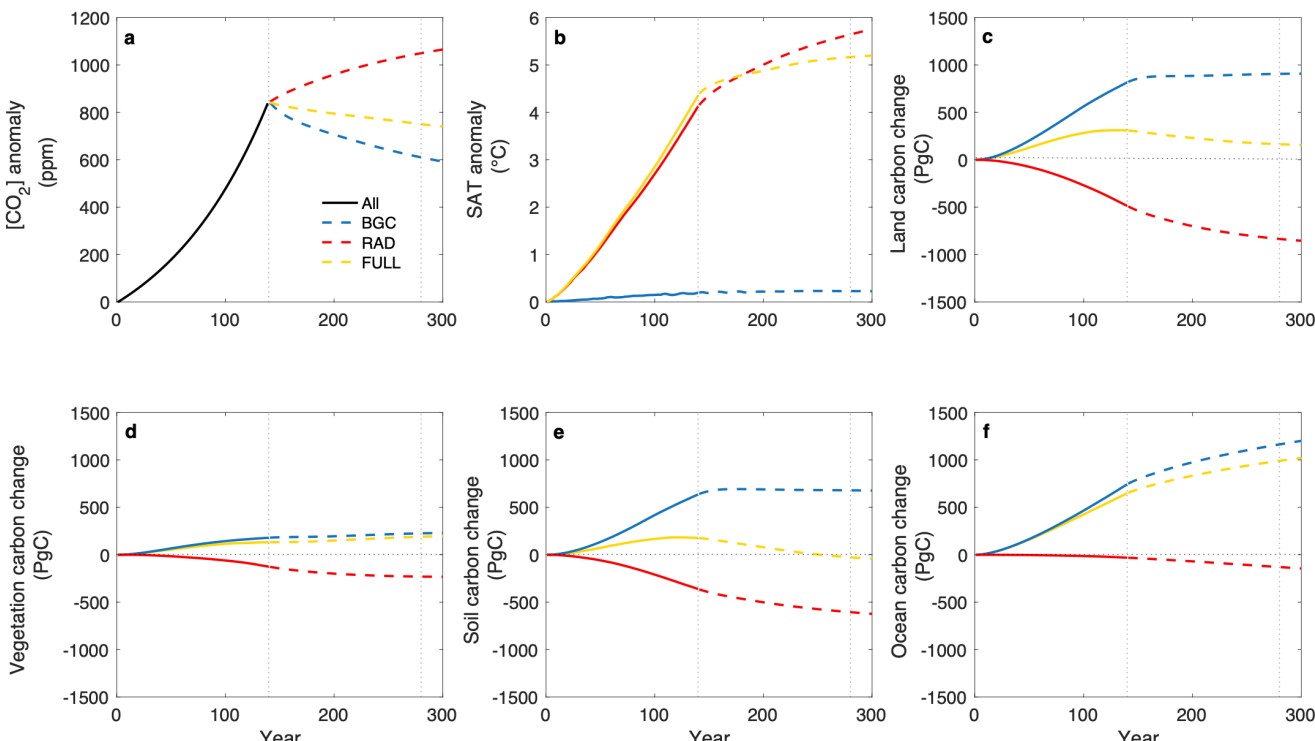

Figure 7: a. Atmospheric $CO_2$ concentration anomaly b. surface air temperature anomaly c. land carbon change d. vegetation carbon change e. soil carbon change and f. ocean carbon change for the zero emissions simulations relative to 1850 (preindustrial). ALL = the CDR-reversibility ramp-up phase from which all modes are initialized; BGC = biogeochemically coupled, RAD = radiatively coupled and FULL = fully coupled. Solid lines are for the ramp-up phase; dashed lines are for the zero emissions phase.

### 3.2.2 "Inertia Corrected" Approach: Isolating the Response to Negative Emissions

The "inertia corrected" approach uses the zero emissions simulations described in the previous section to isolate the response to negative emissions in the "CDR-reversibility" simulations by taking the difference between the ramp-down phase of the RAD (BGC) "CDR-reversibility" simulation and the RAD (BGC) zero emissions simulation. In the BGC mode, despite our attempt to reduce climate system inertia in our novel approach, carbon pools do not return to their preindustrial states at the time atmospheric $CO_2$ returns to preindustrial levels **(figure 5)**. In the RAD mode, all carbon pools eventually gain more
carbon than they held at preindustrial **(figure 6)**.

The "inertia corrected" approach removes the initial carbon increase in the "CDR-reversibility" BGC mode **(figure 5)** and removes the initial carbon decrease in the "CDR-reversibility" RAD mode **(figure 6)** reducing the width of the hysteresis. Zickfeld et al. (2016) used zero emissions to isolate the response to negative emissions and observed a reduction in the initial
carbon change at the beginning of the ramp-down phase consistent with our results. In our approach, the hysteresis may persist because of the different configurations in which the "CDR-reversibility" and "zeroemit" simulations were run, that is, that the former were run with prescribed atmospheric $CO_2$ concentration, whereas the latter were emissions-driven, may also impact the quantification of the inertia. Another possibility may be irreversible changes in vegetation distribution in the "CDR-reversibility" ramp-down phase that are caused by state changes rather than inertia. When the $CO_2$ decrease is prescribed, the
earth system is in a state of elevated $CO_2$ concentration and surface air temperature, which may lead to a different vegetation response than to an equivalent $CO_2$ increase applied from a preindustrial state (Zickfeld et al., 2021). Alternatively, the remaining hysteresis may show that the linearity assumption made in this experiment is not satisfied; the linearity assumption made here is that the total carbon cycle response in the ramp-down phase is a linear combination of the committed response following increasing $CO_2$ concentration and temperature, and the response driven by the decrease in atmospheric $CO_2$ and
temperature in the ramp-down phase (**see Section 2.4.1:** Eq. [8])

After isolating the response to negative emissions alone in the "inertia corrected" approach, the magnitudes of $\beta_L$ and $\beta_O$ are smaller in the ramp-down phase as compared to their respective magnitudes in the ramp-up phase, but the magnitudes of $\gamma_L$ and $\gamma_O$ become larger in the ramp-down phase **(Table 1)**. In the ramp-down phase, the magnitudes of $\beta$ and $\gamma$ from our novel
approach are larger compared to those from the "CDR-reversibility" simulation, implying greater land and ocean carbon loss due to changes in $CO_2$ concentration alone and greater land and ocean carbon gain due to changes in climate alone. For example, a decrease in atmospheric $CO_2$ of one ppm would result in the loss of 0.68 PgC of land carbon in the standard approach and 0.80 PgC of land carbon in our approach due to changes in $CO_2$ concentration alone, whereas, cooling by one degree, would result in land carbon gain of 56.4 PgC in the standard approach and almost three times as much (157.1 PgC) in
our approach due to changes in climate alone.

## 4 Discussion

Our results from the "CDR-reversibility" simulation show that, due to changes in $CO_2$ concentration alone, carbon pools take up carbon in the ramp-up phase, continue to take up carbon in the early ramp-down phase, then switch into sources of carbon. Due to changes in climate alone, carbon pools lose carbon in the ramp-up phase, continue to lose carbon in the ramp-down phase, then switch into carbon sinks. Furthermore, the land and ocean carbon pools do not return to their preindustrial states at the end of both modes, suggesting that land and ocean carbon changes in the ramp-up phase are irreversible on centennial timescales. The differences in the magnitudes of carbon cycle feedbacks in the ramp-up and ramp-down phases, as quantified by feedback parameters, are likely largely due to climate system inertia. This inertia generally reduces the magnitude of both feedbacks in the ramp-down phase (under negative emissions) relative to feedbacks in the ramp-up phase (under positive emissions), implying reduced land and ocean carbon loss due to changes in $CO_2$ concentration alone and reduced land carbon gain due to the changes in climate. The exception is the ocean that continues to lose carbon in the ramp-down phase, implying increased carbon loss due to changes in climate alone.

To quantify the carbon cycle inertia, that is, the response to the prior increasing $CO_2$ trajectory, we ran zero emissions simulations in fully coupled, biogeochemically coupled and radiatively coupled modes. Consistent with previous studies, the ocean continues to sequester carbon in the fully coupled zero emissions simulation (MacDougall et al., 2020). The terrestrial biosphere switches into a carbon source after emissions cease. Carbon uptake, largely by the ocean sink, decreases the atmospheric $CO_2$ concentration. Surface air temperature increases due to the interplay between declining $CO_2$ concentration and ocean heat uptake (Matthews & Caldeira, 2008; Solomon et al., 2009; Arora et al., 2013). While the carbon cycle response is consistent with the behaviour of the UVic ESCM in the Zero Emissions Commitment Model Intercomparison Project (ZECMIP) (MacDougall et al., 2020), the UVic ESCM response in ZECMIP is noticeably different from the rest of the Earth system models. On centennial times, the UVic ESCM is the only model with a positive zero emissions commitment. However, most of the other models do not represent permafrost carbon. The carbon pools in the biogeochemically coupled and radiative coupled zero emissions simulations also exhibit inertia: the land and ocean carbon pools continue to grow after cessation of emissions in the biogeochemically coupled simulation, whereas both carbon pools reduce in the radiatively coupled simulation.

Assuming linearity in the response to increasing and decreasing $CO_2$ concentrations (**see Section 2.4.1:** Eq. [8]), we subtract the zero emissions simulations from the "CDR-reversibility" simulations, to isolate the response to negative emissions alone. We find that in the ramp-down phase, the magnitudes of $\beta$ and $\gamma$ from our novel approach are generally larger as compared to those from the "CDR-reversibility" simulation, implying greater land and ocean carbon loss due to changes in $CO_2$ concentration and greater land and ocean carbon gain due to changes in climate if feedback parameters from our approach are applied instead. Furthermore, land and ocean carbon changes in the ramp-up phase remain irreversible in our simulations.

A similar feedback analysis was conducted for ocean carbon cycle feedbacks using the Norwegian Earth System Model (NorESM) (Schwinger & Tjiputra, 2018). Schwinger and Tjiputra calculated ocean concentration-carbon and climate-carbon feedback parameters using the same carbon cycle feedback framework and "CDR-reversibility" simulations used here. Their results also show a lagged ocean carbon response to the prior increasing $CO_2$ trajectory in the ramp-down phase, and as a result, the magnitude of both carbon cycle feedbacks is smaller in the ramp-down phase than in the ramp-up phase.

540

We compare carbon cycle feedback parameters quantified from the "CDR-reversibility" ramp-up phase to model means and standard deviations from CMIP5 and CMIP6 – the fifth and sixth phases of the Coupled Model Intercomparison Project – respectively (Arora et al., 2020) **(see Table S1)**. The concentration-carbon feedback parameter for land ($\beta_L$) is generally consistent with those from CMIP5 and CMIP6, while the ocean concentration-carbon feedback parameter ($\beta_O$) lies slightly above the CMIP6 range (mean ± 1 standard deviation). The land climate-carbon feedback parameter ($\gamma_L$) lies well above the CMIP5 and CMIP6 ranges, implying a stronger sensitivity to warming relative to CMIP5 and CMIP6 models. The ocean climate-carbon feedback parameter ($\gamma_O$) lies slightly above the ranges for CMIP5 and CMIP6. We have included in the supplement feedback parameters at twice the preindustrial $CO_2$ concentration ($2xCO_2$), which are more relevant, in terms of atmospheric $CO_2$ levels and warming, for real-world mitigation scenarios (**Table S2**).

550

We use the UVIC ESCM, an EMIC, due to the number of simulations and length of model integration required in this study. Compared to comprehensive Earth system models, EMICs generally have coarser resolution and represent less Earth system processes at a lower level of detail. Moreover, the version of the UVic ESCM used here does not represent the nitrogen cycle on land and its coupling to the carbon cycle, which has ramifications for the estimated magnitude of carbon cycle feedbacks. Models without a nitrogen cycle exhibit greater land carbon gain under increasing $CO_2$ concentrations relative to other CMIP5 and CMIP6 models, that is, the concentration-carbon feedback parameter is more positive (**Table S1**). They also exhibit greater carbon loss under increasing $CO_2$ concentrations, that is, the climate-carbon feedback parameter is more negative. Therefore, the magnitude of both carbon cycle feedbacks in this study is generally larger under increasing $CO_2$ concentrations relative to other CMIP5 and CMIP6 models with a nitrogen cycle. Due to the exclusion of the nitrogen cycle, the UVic ESCM is expected to exhibit greater land carbon gain due to changes in climate alone under decreasing $CO_2$ concentrations relative to CMIP5 and CMIP6 models with a nitrogen cycle. Nitrogen mineralization will likely decline as surface air temperature declines, reducing land carbon gain due to changes in climate alone in a model with the nitrogen cycle. The direction of land carbon change due to changes in $CO_2$ concentration alone is less certain. With the consideration of nitrogen limitation, the already weakened $CO_2$ fertilization effect under declining $CO_2$ concentrations could be further constrained, exacerbating the carbon loss due to changes in $CO_2$ concentration alone. However, this may be counteracted by an enhanced rate of photosynthesis as declining $CO_2$ concentrations decrease carbon-nitrogen ratios.

Each of the two approaches used here to quantify carbon cycle feedback parameters has its benefits and drawbacks. Because the "CDR-reversibility" simulation is commonly used in literature (Schwinger & Tjiputra, 2018; Keller et al., 2018; Zickfeld et al., 2016), it allows easier comparison of results across models. However, research shows that this idealized scenario may delay the land sink-to-source transition, and underestimate ocean carbon uptake and the strength of the permafrost carbon feedback (MacDougall, 2019). Furthermore, this scenario requires a period of high positive emissions followed immediately by a period of high negative emissions. The yearly rate of increase in atmospheric $CO_2$ concentration (1%/yr) in the ramp-up phase is twice the rate inferred from historical data (MacDougall, 2019) and achieving such a strong peak and decline is highly unlikely given the scale of negative emissions technologies required.

In their 2016 paper, Zickfeld et al. used zero emissions simulations to correct for the thermal and carbon cycle inertia in a suite of "CDR-reversibility" simulations, similar to our novel approach in this study. This reduced, but did not eliminate the climate system inertia, consistent with our results. Although our approach does not eliminate the inertia, it provides a more accurate estimate of the magnitude of carbon cycle feedbacks in the ramp-down phase by reducing the response to the prior $CO_2$ trajectory, bringing the estimate closer to a quantification of carbon cycle feedbacks under negative emissions alone. The remaining inertia may be associated with the different configurations in which the "CDR-reversibility" and "zeroemit" simulations were run: the former were run in concentration-driven mode whereas, the latter were emissions-driven. Therefore, changes in land and ocean carbon fluxes affect the atmospheric $CO_2$ concentration in the zero emissions simulations, but not in the "CDR-reversibility" simulations. Alternatively, the remaining inertia may be related to irreversible changes in vegetation distribution in the "CDR-reversibility" simulations. Lastly, the linearity assumption made in this experimental design may not hold, that is, the total carbon cycle response in the ramp-down phase may not be a linear combination of the committed response following increasing $CO_2$ concentration and temperature, and the response driven by the decrease in atmospheric $CO_2$ and temperature in the ramp-down phase. If the responses to increasing and decreasing $CO_2$ concentrations are not additive, then the zero emissions simulations may not quantify and remove all the inertia in the "CDR-reversibility" simulations.

**5 Conclusion**

Carbon cycle feedbacks under negative emissions have previously been quantified from the ramp-down phase of the "CDR-reversibility" simulation. However, this approach underestimates the magnitudes of carbon cycle feedbacks because the response in the ramp-down phase includes climate system inertia effects that generally weaken both feedbacks. Our novel approach aims to reduce the inertia in the ramp-down phase, thereby improving the quantification of carbon cycle feedbacks under negative emissions. We find that the magnitudes of the concentration-carbon and climate-carbon feedbacks under negative emissions are larger in our approach as compared to the standard approach. The concentration-carbon feedback drives greater land and ocean carbon release under negative emissions in our approach than in the standard approach. The climate-

carbon feedback promotes more land and ocean carbon sequestration in our approach than in the standard approach. This has two implications: using feedback parameters from the standard approach will (**1**) underestimate land and ocean carbon release under negative emissions due to changes in $CO_2$ concentration alone (concentration-carbon feedback), and (**2**) underestimate land and ocean carbon gain due to changes in climate alone (climate-carbon feedback). Given that the concentration-carbon feedback is the dominant feedback, quantifying carbon cycle feedbacks under negative emissions from the "CDR-reversibility" simulation will result in the underestimation of carbon loss under negative emissions, thereby overestimating the effectiveness of negative emissions in drawing down $CO_2$.

Future research should test the robustness of these results in a multi-model framework. A first step could be analyzing the "CDR-reversibility" simulations in three modes (biogeochemically coupled, radiatively coupled and fully coupled) in the next CMIP phase. In addition, increasing and decreasing $CO_2$ trajectories could be applied from an equilibrium state to overcome issues related to climate system inertia.

## 5 Code/Data Availability

The UVic ESCM data is stored at https://doi.org/10.20383/102.0732 and the model code for UVic ESCM 2.10 is available at http://terra.seos.uvic.ca/model/2.10/.

## 6 Author contribution

K.Z. developed the research question and worked with C.N. on the initial data analysis. V.R.C ran the model simulations and worked with K.Z. to analyse and interpret the model data and write the manuscript. C.N. also helped revise the manuscript.

## 7 Competing interests

The authors declare no competing interests.

## 8 Acknowledgements

This research was funded by the Natural Sciences and Engineering Research Council (NSERC) Discovery Grant Program. Computing resources were provided by the Digital Research Alliance of Canada (formerly Compute Canada).

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
