# Peer review of "Quantifying land carbon cycle feedbacks under negative CO2 emissions"

_Biogeosciences, 2022_

## Referee Comment (RC1)

**Review for *Quantifying land carbon cycle feedbacks under negative CO2 emissions* by Chimuka et al., submitted to Biogeosciences**

The authors present a new approach to quantify the land carbon cycle feedback under negative CO2 emission. The UVic ESCM, an Earth system model of intermediate complexity (EMIC), is utilized to conduct the CDR-reversibility experiment, where the model is driven by 1-percent ramp-up and ramp-down of atmospheric CO2 concentration. With the C4MIP-type setup of BGC and RAD, where only the biogeochemical and radiation effects are included in the model in order to separate the CO2 concentration effect and climate effect, the carbon cycle feedback parameters of land and ocean are quantified for the ramp-up and ramp-down phases respectively. The authors further conduct the emission-driven Zeroemit experiments, which stop the emission and have the carbon cycle freely evolving. The results are again used to calculate the feedback parameters. By comparing the feedback parameters calculated from the ramp-down phase and Zeroemit experiments, the effects of climate inertia are isolated, and the resulting feedback parameters of negative emissions are then closer to that of positive emissions.

The manuscript is well-written and clearly structured. The results are also nicely presented. There are only some issues remain to be clarified in the manuscript. Please see below for my comments.

1. I have some issues with the terminologies used in the manuscript. For example, in the first research questions raised by the authors, the magnitudes of carbon cycle feedbacks under negative and positive emissions are to be compared. However, it is answered in the manuscript that the feedbacks are different because of the climate inertia after the ramp-up phase. However, under a paleoclimate or future climate change context, negative emission does not necessarily immediately follow a ramp-up phase as in the CDR-reversibility experiments.

2. While the results of the current study is helpful for understanding the climate system, it would be better if implications can be drawn connected to current climate change and possible future scenarios corresponding to our climate targets.

3. The authors are encouraged to further connect the results of the current study more to the context of some of the following studies:

   - Jeltsch-Thömmes, A., Stocker, T. F., & Joos, F. (2020). Hysteresis of the Earth system under positive and negative CO 2 emissions. Environmental Research Letters, 15(12), 124026. https://doi.org/10.1088/1748-9326/abc4af
   - Koven, C. D., Arora, V. K., Cadule, P., Fisher, R. A., Jones, C. D., Lawrence, D. M.,

Lewis, J., Lindsay, K., Mathesius, S., Meinshausen, M., Mills, M., Nicholls, Z., Sanderson, B. M., Séférian, R., Swart, N. C., Wieder, W. R., and Zickfeld, K.: Multi-century dynamics of the climate and carbon cycle under both high and net negative emissions scenarios, Earth Syst. Dynam., 13, 885–909, https://doi.org/10.5194/esd-13-885-2022, 2022.

- MacDougall, A. H.: Estimated effect of the permafrost carbon feedback on the zero emissions commitment to climate change, Biogeosciences, 18, 4937–4952, https://doi.org/10.5194/bg-18-4937-2021, 2021.

Minor comments:

- The authors are encouraged to provide some insights of what the differences might be between using a comprehensive Earth system model and an EMIC as UVic.
- L12: UVic is not an Earth system model. I would prefer to always specify out that UVic is an EMIC.
- L13-L14: The carbon cycle feedbacks differ in ramp-up and ramp-down phases, not because the difference between positive and negative emission, but because the climate inertia, as mentioned in the manuscript.
- L125-L129: How long is the Zeroemit simulation? Additionally, it is not mentioned in the manuscript at which time point the feedback parameters are calculated.
- L301-L305: I would expect at which time point the feedback parameters are calculated should already be presented in Section 2.
- L353:
    - Figure 7 caption: Should be *(e)* soil carbon change and *(f)* ocean carbon change.
    - The meaning of *All* is not explained.

- L368-L371: The sentence could be rewritten to made simpler.

---

## Author Comment (AC1)

**Response to Anonymous Referee #1**

The authors present a new approach to quantify the land carbon cycle feedback under negative CO2 emission. The UVic ESCM, an Earth system model of intermediate complexity (EMIC), is utilized to conduct the CDR-reversibility experiment, where the model is driven by 1-percent ramp-up and ramp-down of atmospheric CO2 concentration. With the C4MIP-type setup of BGC and RAD, where only the biogeochemical and radiation effects are included in the model in order to separate the CO2 concentration effect and climate effect, the carbon cycle feedback parameters of land and ocean are quantified for the ramp-up and ramp-down phases respectively. The authors further conduct the emission-driven Zeroemit experiments, which stop the emission and have the carbon cycle freely evolving. The results are again used to calculate the feedback parameters. By comparing the feedback parameters calculated from the ramp-down phase and Zeroemit experiments, the effects of climate inertia are isolated, and the resulting feedback parameters of negative emissions are then closer to that of positive emissions.

The manuscript is well-written and clearly structured. The results are also nicely presented.

We thank the reviewer for taking time to review our manuscript, and for their positive feedback.

There are only some issues remain to be clarified in the manuscript. Please see below for my comments.

- 1. I have some issues with the terminologies used in the manuscript. For example, in the first research questions raised by the authors, the magnitudes of carbon cycle feedbacks under negative and positive emissions are to be compared. However, it is answered in the manuscript that the feedbacks are different because of the climate inertia after the ramp-up phase. However, under a paleoclimate or future climate change context, negative emission does not necessarily immediately follow a ramp-up phase as in the CDR-reversibility experiments.
  - We thank the reviewer for their comments. We recognize that in the real world, • negative emissions are unlikely to follow a ramp-up phase. In future emissions scenarios consistent with our climate targets, the ramp-up phase is followed by a zero emissions phase (Rogelj et al., 2018). Future emissions scenarios with netnegative emissions typically include several different phases (Rogelj et al., 2018), all of which elicit different system responses (Jones et al., 2016; MacDougall et al., 2020). For example, the first Shared Socioeconomic Pathway (SSP1) includes (1) a positive emissions phase with increasing emissions, (2) a net-positive emissions phase with decreasing emissions (possibly with carbon removals compensating some positive emissions), (3) a zero emissions phase, and (4) a netnegative emissions phase (Rogelj et al., 2018). Understanding carbon cycle feedbacks under negative emissions directly from these scenarios would be difficult because climate system inertia will likely make it difficult to disentangle the responses to each phase. As a result, we selected an idealized simulation that allows us to independently analyze the response to negative emissions. Our approach is similar to that taken in CMIP6; the 1%/yr scenario may be idealized,

but its simplicity allows for better understanding of carbon cycle feedbacks without confounding model-related factors (Arora et al., 2020).

**References**

Arora, V. K., Katavouta, A., Williams, R. G., Jones, C. D., Brovkin, V., Friedlingstein, P. ... Ziehn, T. (2020). Carbon-concentration and carbon-climate feedbacks in CMIP6 models, and their comparison to CMIP5 models. Biogeosciences, 17, 4173-4222. https://doi.org/10.5194/bg-17-4173-2020

Jones, C. D., Ciais, J., Davis, S. J., Friedlingstein, P., Gasser, T., Peters, G. P. ... Wiltshire, A. (2016). Simulating the Earth System response to negative emissions. Environ. Res. Lett., 11, 095012.

MacDougall, A. H., Frölicher, T. L., Jones, C. D., Rogelj, J., Matthews, H. D., Zickfeld, K. ... Ziehn, T. (2020). Is there warming in the pipeline? A multi-model analysis of zero emissions commitment of  $CO_2$ . Biogeosciences, 17, 2987 – 3016. https://doi.org/10.5194/bg-17-2987-2020

Rogelj, J., Popp, A., Calvin, K. V., Luderer, G., Emmerling, J., Gernaat, D., ... Tavoni, M. (2018). Scenarios towards limiting global mean temperature increase below 1.5 °C. Nature Climate Change, 8(2), 325–332. https://doi.org/10.1038/s41558-017-0064-y

- 2. While the results of the current study is helpful for understanding the climate system, it would be better if implications can be drawn connected to current climate change and possible future scenarios corresponding to our climate targets.
  - We agree that the next step should be to quantify carbon cycle feedbacks in policy-relevant scenarios. Here we use the CDR-reversibility scenario for the methodological reasons given in the response to the previous comment, and for consistency with the literature on carbon cycle feedbacks under positive emissions which uses the 1%/year scenario (Arora et al., 2020). We will include in the supplement feedback parameters at twice the preindustrial CO2 concentration (2xCO2), which are more relevant, in terms of atmospheric CO2 levels and warming, for real-world mitigation scenarios.

**References**

Arora, V. K., Katavouta, A., Williams, R. G., Jones, C. D., Brovkin, V., Friedlingstein, P. ... Ziehn, T. (2020). Carbon-concentration and carbon-climate feedbacks in CMIP6 models, and their comparison to CMIP5 models. Biogeosciences, 17, 4173-4222. https://doi.org/10.5194/bg-17-4173-2020

- 3. The authors are encouraged to further connect the results of the current study more to the context of some of the following studies:
  - a. Jeltsch-Thömmes, A., Stocker, T. F., & Joos, F. (2020). Hysteresis of the Earth system under positive and negative CO2 emissions. Environmental Research Letters, 15(12), 124026. https://doi.org/10.1088/1748-9326/abc4af
  - b. Koven, C. D., Arora, V. K., Cadule, P., Fisher, R. A., Jones, C. D., Lawrence, D. M., Lewis, J., Lindsay, K., Mathesius, S., Meinshausen, M., Mills, M., Nicholls, Z., Sanderson, B. M., Séférian, R., Swart, N. C., Wieder, W. R., and Zickfeld, K.: Multicentury dynamics of the climate and carbon cycle under both high and net negative emissions scenarios, Earth Syst. Dynam., 13, 885–909, https://doi.org/10.5194/esd-13-885-2022, 2022.
  - c. MacDougall, A. H.: Estimated effect of the permafrost carbon feedback on the zero emissions commitment to climate change, Biogeosciences, 18, 4937–4952, https://doi.org/10.5194/bg-18-4937-2021, 2021.

We thank the reviewer for these literature suggestions. We will assess the relevance of each paper to our study and update our background and discussion sections accordingly.

- 4. Minor comments:
  - a. The authors are encouraged to provide some insights of what the differences might be between using a comprehensive Earth system model and an EMIC as UVic.

We will discuss this briefly in the discussion section.

b. L12: UVic is not an Earth system model. I would prefer to always specify out that UVic is an EMIC.

Done.

c. L13-L14: The carbon cycle feedbacks differ in ramp-up and ramp-down phases, not because the difference between positive and negative emission, but because the climate inertia, as mentioned in the manuscript.

We have rephrased those sentences to make this more clear.

- d. L125-L129: How long is the Zeroemit simulation? Additionally, it is not mentioned in the manuscript at which time point the feedback parameters are calculated.
- e. We thank the reviewer for pointing this out. The Zeroemit simulation is 500 years long, and we take the difference between the negative emissions phase (year 141 280) and the first 140 years of the Zeroemit simulation for each of the biogeochemically and radiatively coupled simulations. We have now included this

information in Section 2.2: Model Simulations and Section 2.3: Approaches to Feedback Quantification.

- f. L301-L305: I would expect at which time point the feedback parameters are calculated should already be presented in Section 2.
- g. We agree. We have moved that paragraph to Section 2.3: Approaches to Feedback Quantification.
- h. L353: Figure 7 caption: Should be (e) soil carbon change and (f) ocean carbon change. The meaning of All is not explained.

The "ALL" label refers to the fact that all three modes (fully coupled, biogeochemically coupled and radiatively coupled) are initialized from the same simulation: the ramp-up phase of the CDR-reversibility simulation. We have clarified this in the figure caption.

i. L368-L371: The sentence could be rewritten to made simpler

We thank the reviewer for pointing this out. We will rephrase this sentence.

---

## Author Comment (AC2)

Response to Community Reviewer: Irina Melnikova (with the inputs of co-authors of Melnikova et al. (2021))

The authors explore carbon cycle feedbacks under an idealized 1%CO2-CDR overshoot scenario using an intermediate complexity model UVic ESCM and introduce a novel approach that uses zero emissions simulations to reduce the climate system inertia when quantifying feedback parameters during the ramp-down period.

I and other co-authors of a closely-related study (Melnikova et al., 2021, hereafter M21) would like to draw the authors' attention to our study as it may have been overlooked when the authors say:
L85: "Our study complements the only existing study on ocean carbon cycle feedbacks under negative emissions (Schwinger & Tjiputra, 2018) by exploring the behaviour of these feedbacks on land."

- *We thank the reviewers for bringing this to our attention. We will include this paper in our background section, and potentially, our discussion section based on relevance to our results.*

It would be interesting to see a comparison of the analysis of the carbon cycle feedbacks under the idealized 1%CO2-CDR scenario with SSP5-3.4-OS scenario, and I would be pleased to provide the data if the authors are interested.

- *We thank the reviewers for sharing SSP5 data with us. Indeed, an analysis of that nature would be interesting. For this paper, however, our goal is to compare magnitudes of carbon cycle feedbacks under positive and negative emissions, and we prefer to focus this on an idealized scenario that allows us to separately quantify carbon cycle feedbacks under positive and negative $CO_2$ emissions.*

Particularly, in M21 (section "4.2. The Peaks of Land and Ocean Carbon Uptakes"), we discuss the balance between GPP and TER that could be useful for the proposed analysis by the authors on balance between NPP and soil respiration.

- *We will review that section and include any insights we gain to our paper.*

Most importantly, the conclusions of this new study sound somewhat opposite to the conclusions of M21 where we stated that: "The carbon cycle feedback parameters amplify after the CO2 concentration and temperature peaks … so that land and ocean absorb more carbon per unit change in the atmospheric CO2 change (stronger negative feedback) and lose more carbon per unit temperature change (stronger positive feedback) compared to if the feedbacks stayed unchanged". In contrast, the study by Chimuka et al. concludes on a "reduced carbon loss due to the concentration-carbon feedback and reduced carbon gain due to the climate-carbon feedback."

I am curious about what drove the discrepancy in the conclusions and encourage the

authors to add some discussion that could be useful for the scientific community and could prevent any confusion about the conclusions.

While I am not sure for the reasons that drove the discrepancy, I speculate it could be (i) the methodology used to calculate the feedback parameters (i.e., in this study, "Feedbacks under negative emissions are computed at the return to preindustrial levels (end of ramp-down phase) using changes in carbon pools, atmospheric CO2 concentration, and surface air temperature computed relative to the time of peak atmospheric CO2", while M21 computed them relative to piControl). In fact, for our M21 analysis we considered using (1) piControl, (2) time of CO2 and temperature peaks, and (3) "new equilibrium state" at the end of the simulation. However, we chose (1) because using (2) in the more "realistic" SSP scenario would result in too small values of ΔCO2 and temperature during most part of the ramp-down phase, resulting in ill-defined quantities. Besides, UVic ESCM shows no lag between the peaks of CO2 concentration and global surface temperature but it is not the case in some of the more complex models (e.g., Boucher et al., 2012, shows a lag of temperature peak over the ocean; in M21, the lag of temperature peak is up to 30 years, depending on the ESM). The discrepancy in conclusions of the two studies could also be due to (ii) the proposed method to remove the impact of climate inertia by using additional zero-emission simulations. Finally, (iii) the discrepancy could root in the difference between the idealized 1%CO2-CDR and SSP5-3.4-OS scenarios (e.g., due to scenario dependency of feedback parameters). I suggest adding discussion on this matter, especially in terms of the implications of translating the conclusions from the idealized scenarios to the more socially-relevant ones.

- *We thank the reviewers for their comments. Indeed, the differences in our conclusions could cause confusion if not clarified. The main reason for the discrepancy in our view is the methodology, as the reviewers have stated. We chose to compute our feedback parameters under negative emissions relative to the time of peak $CO_2$ concentration. We initially considered computing the feedbacks relative to preindustrial (closer to the approach used in M21), but encountered an issue with the concentration-carbon feedback parameter. The concentration-carbon feedback parameter (β) is computed as:*

$$[3.3.1] \quad \beta_L = \frac{\Delta C_L}{\Delta C_A} \qquad\qquad [3.4.1] \quad \beta_O = \frac{\Delta C_O}{\Delta C_A}$$

  *Under negative emissions, the $CO_2$ concentration declines, making the denominator smaller and smaller, leading to β showing exponential behaviour. Therefore, towards the end of the ramp-up phase, β becomes less of a representation of the land and ocean sensitivity to $CO_2$ changes, as this signal becomes obscured by the exponential behaviour. We found β under negative emissions to be more meaningful when computed relative to the time of peak $CO_2$ concentration.*
- *We would also like to clarify that there are two separate conclusions based on our two research questions:*
  - *In the first approach, we compare the magnitudes of carbon cycle feedbacks under positive and negative emissions, and find that **both feedback parameters***

*are smaller under negative emissions* due to climate system inertia. Here is where we conclude that, with climate system inertia effects, there will be reduced carbon loss due to the concentration-carbon feedback and reduced carbon gain due to the climate-carbon feedback.

- ▪ *If we compute carbon cycle feedbacks under negative emissions relative to preindustrial (closer to the approach used in M21), we find that the magnitude of both feedbacks is larger in the ramp-down phase, consistent with the findings in M21. The caveat here, as mentioned before, is that the magnitude of the concentration-carbon feedback parameter becomes increasingly driven by exponential behaviour as the $CO_2$ concentration declines.*

- o *We then proceed in the second approach, to adjust the magnitude of carbon cycle feedbacks under negative emissions by isolating the response to negative emissions alone, so that they can be more comparable to those under positive emissions. Here, we find that the **concentration-carbon feedback parameter is still smaller under negative emissions as compared to positive emissions**, but now larger than before the correction was done. **The climate-carbon feedback parameter is larger under negative emissions than under positive emissions**, and is also now larger than before the correction was done. Here, is where we conclude that, using uncorrected feedback parameters could be risky because it results in an underestimation of carbon loss under negative emissions and an overestimation of the carbon gain, and given that the latter feedback is more dominant, an overall underestimation of carbon loss under negative emissions and an overestimation of the effectiveness of negative emissions.*

- • *We agree that the next step should be to quantify carbon cycle feedbacks in policy-relevant scenarios. Here we use the CDR-reversibility scenario for the methodological reasons given in the response to the previous comment, and for consistency with the literature on carbon cycle feedbacks under positive emissions which uses the 1%/year scenario (Arora et al., 2020). We will include in the supplement feedback parameters at twice the preindustrial $CO_2$ concentration ($2xCO_2$), which are more relevant, in terms of atmospheric $CO_2$ levels and warming, for real-world mitigation scenarios.*

  *References*

  *Arora, V. K., Katavouta, A., Williams, R. G., Jones, C. D., Brovkin, V., Friedlingstein, P. ...Ziehn, T. (2020). Carbon-concentration and carbon-climate feedbacks in CMIP6 models, and their comparison to CMIP5 models. Biogeosciences, 17, 4173-4222. https://doi.org/10.5194/bg-17-4173-2020*

Other comments

1. L341: "Surface air temperature remains relatively constant in the BGC mode. In the FULL mode, the land switches into a source of carbon after missions cease, consistent with the behaviour of the UVic ESCM in the Zero Emissions Commitment Model Intercomparison Project (ZECMIP)"

Yes, but there is a variety of responses among models in ZECMIP. The UVic's behavior in ZECMIP is somewhat different from the majority of models (see figures 2.d and 3.a of MacDougall et al 2020). Could some discussion be added?

*We agree that the UVic response is different from most other models in ZECMIP. We do not treat this here because analyzing the reasons for differences in ZEC between models is beyond the scope of our paper, but we will include a few sentences stating this difference in our discussion section.*

2. Also, we would appreciate seeing a comparison of the 'standard' $\beta$ and $\gamma$ (under 1%CO2 experiments) by UVic to the CMIP6 ensemble in a table or figure to get a better idea of where this version of UVic stands.

*We have included a comparison of the feedback parameters we computed to those from CMIP5 and CMIP6 in Table S1 and included a discussion in Lines 417-424.*

3. L426: "Models without a nitrogen cycle exhibit greater land carbon gain under positive emissions relative to other CMIP5 and CMIP6 models, that is, the concentration-carbon feedback parameter is more positive (Table S2). They also exhibit greater carbon loss under positive emissions, that is, the climate-carbon feedback parameter is more negative."

I am concerned that the authors ignore that the climate-carbon feedback may be both positive (i.e., amplifying climate change) and negative in the colder regions.

*We thank the reviewers for their comment. We would like to first to clarify that we compute carbon cycle feedbacks at the global scale, and therefore, the magnitudes and signs are for the overall feedback. In addition, the signs we refer to in our paper are not the signs of the feedback i.e., negative (positive) feedback parameter ≠ negative (positive) feedback; they are instead the signs of the feedback parameters, which are generally opposite to the sign of the feedback because our feedback parameters are from the perspective of the land and ocean (see **Table 1**). We will clarify this in the text.*

4. L12: "This study investigates land carbon cycle feedbacks under positive and negative CO2 emissions using an Earth system model"

The fact that UVic is not an ESM but EMIC should be made clear throughout the manuscript

*We have clarified this in the text.*

I hope these comments are useful.

Irina Melnikova, with the inputs of co-authors of Melnikova et al. (2021)

*We thank the reviewers for taking time to review our manuscript and providing such a useful and constructive review.*

---

## Author Comment (AC3)

Response to Anonymous Referee #2

Authors quantify carbon-concentration and carbon-climate feedback for negative emissions for an idealized scenario and compare the magnitude of these feedbacks for the positive emissions part of an idealized scenario. The manuscript is relatively well written and in principle it all makes sense. However, I would suggest improving the manuscript in the following ways.

Please include equations in the main text that should clarify your methodology (where you subtract the effect of zero emissions run on quantities considered during the rampdown phase). If a picture is worth 500 words, an equation is worth at least 200 words.
In the absence of the equations, it is difficult to understand your methodology

- *We thank the reviewer for their comments and positive feedback. We agree that including equations for our proposed approach would make our methodology easier to follow. We will include the equations for our proposed approach in the main text, along with the carbon cycle feedback framework in our supplement.*

Please introduce your sign notation in the beginning and then use it consistently throughout the manuscript. Recall that carbon-concentration feedback is negative from the atmosphere's perspective because it reduces atmospheric CO2 If you use the term "when carbon is gained" then please clarify explicitly which component is gaining carbon - land/ocean or the atmosphere.

Near lines 308-313, I was confused with the sign notation even more because it seems, as you interpret it, sign notation reverses during the ramp-down phase. This needs to be better explained because I am unable to understand why sign notation reversal is needed. If carbon-concentration feedback is negative from an atmosphere's perspective (let's say a value of -1.0 Pg C/ppm) this implies that an increase in atmospheric CO2 concentration will be reduced from its initial amount due to this negative feedback. The corollary of this is that if atmospheric CO2 is reducing then the change in CO2 is negative (say -2 ppm) which when multiplied by -1.0 Pg C/ppm yields +2.0 Pg C implying 2 Pg C is added to the atmosphere. All this makes sense in my mind. So why is reversal of sign notation needed?

- *We thank the reviewer for highlighting this – we will address both comments here. We agree that the concentration-carbon feedback remains a negative feedback under negative emissions. Likewise, the climate-carbon feedback remains a positive feedback. What changes here is **the meaning of the sign of the feedback parameters**. The concentration-carbon feedback parameter is calculated by rearranging equations 3.3 and 3.4 for the land and ocean respectively. Hence, the equations are then expressed as:*

$$[3.3.1] \quad \beta_L = \frac{\Delta C_L}{\Delta C_A} \qquad\qquad [3.4.1] \quad \beta_O = \frac{\Delta C_O}{\Delta C_A}$$

- *Under positive emissions, $CO_2$ concentration increases (positive denominator) and both the land and ocean gain carbon (positive numerator) resulting in a positive feedback parameter for both land and ocean. Under positive emissions, a positive $\beta_L$ and $\beta_O$ is associated with land and ocean carbon gain. Under negative emissions, when both numerator and denominator decrease, $\beta_L$ and $\beta_O$ remain positive as the reviewer correctly stated, but now, a positive feedback parameter is associated with land and ocean carbon loss.*
- *We also agree that the phrase "carbon is gained" is unclear, and we have clarified all instances of "carbon is gained" and "carbon is lost" with where it is gained or lost from.*
- *Lastly, we will add a paragraph introducing the sign convention to the carbon cycle feedback framework in our supplement (along with the two equations above for $\beta$ and those for $\gamma$), which we will be moving to Section 2: Methodology in the main text. We will also clarify the meaning of the sign of the feedback parameters under positive and negative emissions, and hence, the reversal of the sign convention.*

I would also like to note that feedback parameters are most "realistic" or "relevant" when found using FULL and BGC runs. The real world operates like a fully-coupled simulation. For finding feedback parameters in addition to FULL we need a BGC or RAD simulation. Since the carbon-concentration feedback is the dominant feedback perhaps it makes more sense to use the BGC simulation.

- *We agree that feedback parameters are more realistic calculated using the FULL-BGC approach. We include feedback parameters calculated from this approach in Table 1 (shown in parentheses) as well as the relevant land, ocean, vegetation and soil carbon changes in Figure S5 of the supplement.*
- *We find that the feedback parameters from the FULL-BCG approach are qualitatively consistent with those from the RAD approach: the magnitude of the climate-carbon feedback parameters calculated from both approaches is smaller under negative emissions than under positive emissions. Moreover, we find that results are easier to interpret in our proposed approach with feedback parameters computed using the RAD approach rather than the FULL-BGC approach. In the latter case, we would need to subtract the difference between FULL and BGC zero emissions simulations from the difference between the FULL and BGC CDR-reversibility ramp-down phases. This double difference would make it difficult to make sense of the resulting feedback parameters.*

Finally, my last major comment is that when in the real world we do ramp down emissions then, at that point in time, the land and ocean C cycles won't be in equilibrium with the atmospheric CO2. There will be inertia in the real system, and the response of land and ocean at the time will be affected by this inertia. So is the purpose of attempting to correct the feedback parameters for this inertia on the ramp-down side only to compare them with their ramp-up counterparts?

- *Yes, that is correct. Since the feedback parameters under negative emissions include land and ocean responses to both the negative emissions and prior positive emissions, we*

*isolate the response to the negative emissions alone by using zero emissions simulations, bringing the feedback parameters closer to the "true" sensitivity of land and ocean carbon under negative emissions.*

Minor comments

1. I realize the purpose of Figure 1 is to clarify things but for me text for easier to follow. Perhaps you can try to improve Figure 1.

*We will improve Figure 1 or reconsider including it in the main text.*

2. Line 107, "generates permafrost". Please reword this sentence. I think it is incorrect to say "generate permafrost". Permafrost is a state which results from sub-zero temperatures.

*Done.*

3. Lines 145-152 need equations to clarify the methodology used.

*As previously mentioned, we will move the carbon cycle feedback framework (including all equations) in Section A of the supplement to Section 2: Methodology of the main text to clarify Lines 145-152.*

4. Line 172. "This temperature change is driven by biophysical responses to increasing CO2". Please add another sentence of explanation at the end of this sentence for completeness.

*We thank the reviewer for highlighting this. The temperature change in the BGC mode is driven by changes in evaporative fluxes. We have included a sentence explaining this.*

5. Please put a zero line in Figures 3c,d,e,f, and Figures 4a,b.

*We will add the zero line to those figures as well as Figures 6 and 7c-f*

6. Lines 258-261 read " … except in the vegetation carbon pool where the width of the hysteresis increases throughout the simulation (figure 5(c)). The land and ocean carbon pools in the RAD mode also exhibit hysteresis (figure 6). The hysteresis in the land carbon pool is dominated by the soil carbon pool (figure 5(d)), and the width of the hysteresis appears to increase throughout the simulation for all carbon pools except the vegetation carbon, which shows nearly constant hysteresis".

I am confused here. Please reword clearly. Hysteresis is defined as the difference in paths going up and down. Isn't hysteresis zero at the point of turn? With this in mind please reword the above sentences.

*We agree with this definition and agree that hysteresis should be 0 at the point of turn. We have changed the phrase "…the width of the hysteresis appears to increase **throughout the simulation** for all carbon pools …" to "…the width of the hysteresis appears to increase **throughout the ramp-down phase** for all carbon pools …" to make this clearer.*

7.  Lines 273-274 read "The ocean holds only 70PgC less than at preindustrial, but unlike the land carbon pool, a miniscule amount of ocean carbon is regained in the rampdown phase (figure 5d)".

But Figure 5d is the soil C figure. Please refer to the correct figure.

*We thank the reviewer for pointing out this error. We corrected the figure reference.*

Line 308 reads "For positive emissions, feedback parameters are positive (negative) for a gain (loss) of carbon". Please consider not using sentences that use pair of parentheses to note two points. This can get very confusing. Also, please clarify whether the gain or loss is by which component – land/ocean or the atmosphere.

*Done.*

9. Line 309 reads " … resulting in a negative denominator (see supplementary equations 3.3 – 3.6)".
There is no denominator in these equations. I think I know what's implied here but it may not be obvious to other readers.

10. Lines 308 – 313. Please use equations here because the sign convention is becoming confusing.

*We will address both comments 9 and 10 here. As mentioned in our response to the second major comment above, we will add the two equations above for β (along with those for γ) to our carbon cycle feedback framework in our supplement as a visual reference and to help readers understand the sign convention.*

11. Comparison of Figure 5a and S4a shows there's more hysteresis in BGC run than in the FULL run. Can this be explained? Isn't this a good reason to use the FULL simulation to find feedback parameters on the ramp-up and ramp-down portions?

*Although we find the difference in hysteresis between the FULL and BGC modes very interesting, we do not explore or explain this further because reversibility is beyond the scope of our study. In addition, assuming you mean using the FULL-BGC approach for computing the climate-carbon feedback in the two phases, conclusions drawn from feedback parameters computed from the FULL-BGC and RAD approaches are consistent in our study (see response to last major comment).*

12. What does "All" means in Figure 7a legend?

*The "ALL" label refers to the fact that all three modes (fully coupled, biogeochemically coupled and radiatively coupled) are initialized from the same simulation: the ramp-up phase of the CDR-reversibility simulation. We have clarified this in the figure caption.*

13. Zero emissions runs were initialized from the end of ramp-up. What does BGC and RAD mean for these runs? Do the RAD and BGC runs in Figure 7, see and not see temperature change, respectively, relative to end of the ramp-up or relative to the preindustrial state? Please clarify.

*As the RAD and BGC runs are initialized from the end of the ramp-up phase, they see and do not see temperature change relative to the end of the ramp-up. Therefore, in the RAD run, the land and ocean see changes in temperature, but see $CO_2$ concentration fixed at four times the preindustrial $CO_2$ concentration (~1120ppm). In the BGC run, the land and ocean see changes in $CO_2$ concentration, but the radiation code remains fixed at four times the preindustrial $CO_2$ concentration. We will clarify this in the text.*

14. Lines 375-377 read "Under negative emissions, the magnitudes of b[eta] and g[amma] from our novel approach are larger compared to those from the "CDRreversibility" simulation WHEN RAMPING UP (CORRECT?), implying greater carbon loss due to the concentration-carbon feedback and greater carbon gain due to the climatecarbon feedback under negative emissions".
"Greater carbon loss" and "greater carbon gain" for what component – land/ocean or atmosphere?

*Here, we are referring to magnitudes of carbon cycle feedback parameters under negative emissions i.e., in the ramp-down phase. Therefore, we are comparing the feedback parameters from the ramp-down phase of the CDR-reversibility simulation to those from our "ramp-down – zeroemit" approach (the ramp-down of the CDR-reversibility simulation minus zero emissions simulation). We also include an example of this with values of feedback parameters in Lines 377-381 that read, "For example, a decrease in atmospheric CO2 of one ppm would result in the loss of 0.68 PgC of land carbon in the standard approach and 0.80 PgC of land carbon in our approach due to the concentration-carbon feedback whereas, cooling by one 380 degree, would result in land carbon gain of 56.4 PgC in the standard approach and almost three times as much (157.1 PgC) in our approach due to the climate-carbon feedback." Finally, "greater carbon loss" and "greater carbon gain" refers to the land and ocean components; we have clarified this in all instances in the manuscript.*

15. Lines 383-384 read "… due to the concentration-carbon feedback, carbon pools take up carbon in the ramp-up phase, continue to take up carbon in the early ramp-down phase."
Actually, it's the other way around. Carbon pools don't behave according to the feedbacks but rather feedbacks are derived from the behavior of the C pools. Please consider rewording.

16. Next two sentences …
"Due to the climate-carbon feedback, carbon pools lose carbon in the ramp-up phase,

continue to lose carbon in the ramp-down phase, then switch into carbon sinks"

"… suggesting that land and ocean carbon changes due to carbon cycle feedbacks …"
Here too, please consider rewording.

*We thank the reviewer for their comments. We will reword both sentences.*

17. Lines 404-405 read " … we subtract the zero emissions simulations from the "CDR reversibility"
simulations …".
Please use equations to show how.

*As mentioned in the first major comment, we will provide the equations in the main text along with the carbon cycle feedback framework in our supplement.*

18. Lines 427-428 read "… concentration-carbon feedback parameter is more positive (Table S2)".
Please clarify if this is from the land's perspective. Please use a single notation consistently.

*This is from the land's perspective. We say, "Models without a nitrogen cycle exhibit greater **land carbon gain** under positive emissions relative to other CMIP5 and CMIP6 models, that is, the concentration-carbon feedback parameter is more positive (Table S2)." [Lines 426-427].*

19. Lines 428-429 read … "They [i.e. land models with N cycle] also exhibit greater carbon loss under positive emissions, that is, the climate-carbon feedback parameter is more negative".

This seems incorrect. Note that land models with N cycle typically have a smaller absolute magnitude of carbon-climate feedback because increase in temperature promotes vegetation growth due to enhanced N mineralization which somewhat compensates for increased soil C respiratory losses.

*We thank the reviewer for their comment. Here, we are referring to models without a N cycle. Lines 427-429 read, "Models without a nitrogen cycle exhibit greater land carbon gain under positive emissions relative to other CMIP5 and CMIP6 models, that is, the concentration-carbon feedback parameter is more positive (**Table S2**). They also exhibit greater carbon loss under positive emissions, that is, the climate-carbon feedback parameter is more negative."*

20. Lines 433 – 435 read "With the consideration of nitrogen limitation, the already weakened CO2 fertilization effect under declining CO2 concentrations would be further constrained, exacerbating the carbon loss due to the concentration-carbon feedback".
This seems like a bit of speculation. Why would this be? It could be the other way around too. If increasing CO2 causes C:N ratios to increase and constrain photosynthesis, more than the case when the N cycle is not represented, then decreasing CO2 should lower C:N ratio and help vegetation photosynthesize a bit more (compared to when the N cycle is

not represented).
Of course, overall photosynthesis will still be reducing since CO2 is going down but off the top of my head it's difficult for me to imagine the effect of N cycle when CO2 is reducing.

Perhaps is prudent to not speculate.

*We thank the reviewer for bringing this to our attention. We had only considered the impact of nitrogen limitation on the $CO_2$ fertilization effect. We will also consider C:N ratios and instead of making a definitive statement of the effect of the N cycle, we will suggest both as potential responses.*

21. Finally, what is the CDR-reversibility simulation? Does this refer to both the ramp-up and ramp-down portions or just the ramp-down portion? Note that the ramp-up portion already has a standard experiment name i.e. 1pctCO2. Please clarify this in the beginning and then use the correct terminology throughout the rest of the manuscript

*This is correct. The CDR-reversibility simulation is the combination of the ramp-up and ramp-down simulation. We say, "To explore how the magnitude of carbon cycle feedbacks under positive emissions differs from that under negative emissions, we ran the "CDR-reversibility" simulation from the Carbon Dioxide Removal Model Intercomparison Project (CDRMIP) (Keller et al., 2018). Starting from a preindustrial equilibrium state, atmospheric $CO_2$ concentration was prescribed to increase at 1% per year until quadrupling, then decline back to preindustrial levels at the same rate." [Lines 119-122]. We also chose to refer to it as the CDR-reversibility ramp-up phase as opposed to "1pctCO2" as we find it clearer which portion of the CDR-reversibility we are referring to. We will make sure to use consistent terminology throughout the manuscript.*

---

## Referee Report (RR1)

**Quantifying land carbon cycle feedbacks under negative CO$_2$ emissions**

V. Rachel Chimuka[1], Claude-Michel Nzotungicimpaye[1,a] & Kirsten Zickfeld[1]

[1]Department of Geography, Simon Fraser University, Burnaby, BC, V5A 1S6, Canada
5   [a] Now at Department of Geography, Planning and Environment, University of Concordia, Montréal, QC, H3G 1M8, Canada

*Correspondence to*: V. Rachel Chimuka (rchimuka@sfu.ca)

**Abstract.** Land and ocean carbon sinks play a major role in regulating atmospheric CO$_2$ concentration and climate. However, their future efficiency depends on feedbacks in response to changes in atmospheric CO$_2$ concentration and climate, namely the concentration-carbon and climate-carbon feedbacks. Since carbon dioxide removal is a key mitigation measure in emission

10   scenarios consistent with global temperature goals in the Paris agreement, understanding carbon cycle feedbacks under negative CO$_2$ emissions is essential. This study investigates land carbon cycle feedbacks under positive and negative CO$_2$ emissions using an Earth system model of intermediate complexity (EMIC) driven with an idealized scenario of atmospheric CO$_2$ increase and decrease, run in three modes. Our results show that the magnitude of carbon cycle feedbacks differs between the atmospheric CO$_2$ ramp-up and ramp-down phases. These differences are likely largely due to climate system inertia: the

15   response in the ramp-down phase represents the response to both the prior positive emissions and negative emissions. To isolate carbon cycle feedbacks under negative emissions and quantify these feedbacks more accurately, we propose a novel approach that uses zero emissions simulations to reduce this inertia. We find that the magnitudes of the concentration-carbon and climate-carbon feedbacks under negative emissions are larger in our novel approach than in the standard approach. This has two implications: using feedback parameters from the standard approach will (**1**) underestimate land and ocean carbon

20   release under negative emissions due to changes in CO$_2$ concentration alone (concentration-carbon feedback), and (**2**) underestimate land and ocean carbon gain due to changes in climate alone (climate-carbon feedback). Given that the concentration-carbon feedback is the dominant feedback, quantifying carbon cycle feedbacks with the standard approach will result in the underestimation of land and ocean carbon loss under negative emissions, thereby overestimating the effectiveness of negative emissions in drawing down CO$_2$.

**Summary of Comments on bg-2022-168-manuscript-version2_Mar_2023.pdf**

**Page:1**

**Number: 1  Author:  Subject:Highlight  Date:2023-03-14 8:53:25 AM**

This
has two implications: using feedback parameters from the standard approach will (1) underestimate land and ocean carbon
20 release under negative emissions due to changes in $CO_2$ concentration alone (concentration-carbon feedback), and (2)
underestimate land and ocean carbon gain due to changes in climate alone (climate-carbon feedback).

**Author:  Subject:Note  Date:2023-03-14 9:55:05 AM**

I find this confusing to understand. Note that feedback parameters are a way to show the reponse of the system in a simple way to $CO_2$ and T changes. Given their well known scenario-dependence I am not sure if people use them to find land and ocean C gain back from another scenario.

If you are just say that the feeback parameters are different between yours and the standard approach that's sufficient in my view.

**Number: 2  Author:  Subject:Highlight  Date:2023-03-14 8:53:40 AM**

result in the underestimation of land and ocean carbon loss under negative emissions, thereby overestimating the effectiveness
of negative emissions in drawing down $CO_2$.

**Author:  Subject:Note  Date:2023-03-14 8:53:53 AM**

Again same thing can go here.

Can you highlight examples where people use feedback parameters to calculate emissions?

[revised manuscript text omitted]

and climate.

Author:  Subject:Note  Date:2023-03-14 8:54:18 AM

This has to mention radiation too.

**H** Number: 2  Author:  Subject:Highlight  Date:2023-03-14 8:54:25 AM

The radiation module stays fixed at the CO2 level

Author:  Subject:Note  Date:2023-03-14 9:55:38 AM

Please consider rewording this. It's better to say - "radiation module sees  a specified time-invariant CO2
concentration".

**2.4 Carbon Cycle Feedback Framework**

165  We use integrated flux-based feedback parameters (Friedlingstein et al., 2006) to quantify carbon cycle feedbacks in both approaches, under both positive and negative emissions. The total change in land (ocean) carbon is expressed as the sum of two terms: a term representing the change in land (ocean) carbon in response to changes in atmospheric $CO_2$, and a term representing the change in land (ocean) carbon in response to changes in surface air temperature:

170
$$\Delta C_L = \beta_L \Delta C_A + \gamma_L \Delta T \qquad [1]$$
$$\Delta C_O = \beta_O \Delta C_A + \gamma_O \Delta T \qquad [2]$$

The concentration-carbon feedback parameter $\beta$ quantifies the carbon cycle response to changes in $CO_2$ concentration in units of PgC ppm$^{-1}$, whereas the climate-carbon feedback parameter $\gamma$ quantifies the carbon cycle response to changes in climate in
175  units of PgC °C$^{-1}$.

The change in land (ocean) carbon due to the increasing atmospheric $CO_2$ concentration is determined using the biogeochemically coupled simulation. In this simulation, the land and ocean only respond to changes in the $CO_2$ concentration, and therefore, this simulation can be used to quantify the concentration-carbon feedback parameter $\beta$. Warming is still observed
180  in these simulations because the water use efficiency of vegetation increases at higher $CO_2$ concentrations and changes in albedo due to shifts in vegetation structure and spatial distribution, result in a small warming effect (Cox et al., 2004, Boer & Arora, 2013; Arora et al., 2013). However, this warming is considered negligible in this feedback framework. Assuming that $\Delta T = 0$ in Eq. (1) and (2), the change in land (ocean) carbon due to changes in atmospheric $CO_2$ concentration is expressed as:

185
$$\Delta C_L = \beta_L \Delta C_A \qquad [3a]$$
$$\Delta C_O = \beta_O \Delta C_A \qquad [4a]$$

Equations (3a) and (4a) can then be rearranged to solve for the concentration-carbon feedback parameter $\beta$ as follows:

$$\beta_L = \frac{\Delta C_L}{\Delta C_A} \qquad [3b] \qquad \beta_O = \frac{\Delta C_o}{\Delta C_A} \qquad [4b]$$

190
The change in land (ocean) carbon due to climate change is determined using the radiatively coupled simulation. In this simulation, the land and ocean only respond to changes in climate, and therefore, this simulation can be used to quantify the climate-carbon feedback parameter $\gamma$. The change in land (ocean) carbon due to climate change is expressed as:

195

$$\Delta C_L = \gamma_L \Delta T \qquad [5a]$$
$$\Delta C_O = \gamma_O \Delta T \qquad [6a]$$

Equations (5a) and (6a) can then be rearranged to solve for the climate-carbon feedback parameter $\gamma$ as follows:

200

$$\gamma_L = \frac{\Delta C_L}{\Delta T} \qquad [5b] \qquad \gamma_O = \frac{\Delta C_o}{\Delta T} \qquad [6b]$$

An alternative method for quantifying the change in land (ocean) carbon due to climate change uses the fully coupled and biogeochemically coupled simulations (Arora et al., 2013). Here, we refer to this method as the FULL-BGC method. Here, the

205 change in land (ocean) carbon in the biogeochemically coupled simulation (BGC) is subtracted from that in the fully coupled simulation (FC) and expressed as the product of the climate-carbon feedback parameter, and the difference between the surface air temperature changes in the two simulations:

$$\Delta C_L{}^{CLIM} = \Delta C_L{}^{FC} - \Delta C_L{}^{BGC} = \gamma_L(\Delta T^{FC} - \Delta T^{BGC}) \qquad [7]$$

210

$$\Delta C_O{}^{CLIM} = \Delta C_O{}^{FC} - \Delta C_O{}^{BGC} = \gamma_O(\Delta T^{FC} - \Delta T^{BGC}) \qquad [8]$$

The resulting feedback parameters differ from those quantified from the RAD mode (Eq. (5b), (6b)) alone due to nonlinearities in carbon cycle feedbacks (Zickfeld et al., 2011; Schwinger & Tjiputra, 2018).

215 Feedback parameters under positive emissions are computed at the peak atmospheric $CO_2$ concentration (quadruple the preindustrial level) using changes in carbon pools, atmospheric $CO_2$ concentration and surface air temperature computed relative to preindustrial levels. Feedback parameters under negative emissions are computed at the return to preindustrial levels (end of ramp-down phase) using changes in carbon pools, atmospheric $CO_2$ concentration and surface air temperature computed relative to the time of peak atmospheric $CO_2$.

220

Under positive emissions, feedback parameters are positive for land or ocean carbon gain and negative for land or ocean carbon loss. Under negative emissions, however, both atmospheric $CO_2$ concentration and surface air temperature decline, resulting in a negative denominator (see Eq. (3b), (4b), (5b) and (6b)). Therefore, the sign convention is reversed: feedback parameters are negative for a gain in land or ocean carbon (positive numerator divided by negative denominator) and positive for a loss in

225 land or ocean carbon (negative numerator divided by negative denominator). The signs we refer to here, however, are not the signs of the feedback but rather the signs of the feedback parameters, which are generally opposite to the sign of the feedback under positive emissions because our feedback parameters are computed from the perspective of the land and ocean, whereas the sign of the feedback is determined from the perspective of the atmosphere.

[Figure]
 Number: 1  Author:  Subject:Highlight  Date:2023-03-14 8:54:42 AM

Under positive emissions, feedback parameters are positive for land or ocean carbon gain and negative for land or ocean carbon
loss.

[Figure]
 Author:  Subject:Note  Date:2023-03-14 9:56:26 AM

The carbon-cocentration feedback is -ve from the atmosphere's perspective, but positive from the land and ocean's perspective.

Please keep this distinction. Here in this sentence you are talking from feedback parameters for land and ocean.

That is,

beta_A = -(beta_L + beta_O).

I see this is done at the end of this paragraph but please mention this earlier.

[Figure]
 Number: 2  Author:  Subject:Highlight  Date:2023-03-14 8:54:57 AM

Therefore, the sign convention is reversed: feedback parameters
are negative for a gain in land or ocean carbon (positive numerator divided by negative denominator) and positive for a loss in
225 land or ocean carbon (negative numerator divided by negative denominator). The signs we refer to here, however, are not the
signs of the feedback but rather the signs of the feedback parameters, which are generally opposite to the sign of the feedback
under positive emissions because our feedback parameters are computed from the perspective of the land and ocean, whereas
the sign of the feedback is determined from the perspective of the atmosphere.

[Figure]
 Author:  Subject:Note  Date:2023-03-14 8:56:15 AM

This is all very confusing, and the result of expressing change in CO2 concentration related to the peak. If change in CO2 concentration is found relative to PI, then change in CO2 concentration is still +ve until it becomes zero when CO2 gets down to its PI value again.

Please consider making this clear that the sign reversal is the result of how you are starting at the peak and not at the PI value.

**2.4.1 Isolating the Response to Negative Emissions (Ramp-down – Zeroemit Approach)**

230 [1] When negative emissions are applied from a transient (time-evolving) state, the land and ocean respond to both the negative emissions and the prior emissions trajectory (Zickfeld et al., 2016). The land and ocean responses can, therefore, be expressed as the response to negative emissions plus an inertia term that represents the committed response to past history:

$$\Delta C_L = \Delta C_L^{NE} + \Delta C_L^{INERTIA} \quad [9]$$

235
$$\Delta C_O = \Delta C_O^{NE} + \Delta C_O^{INERTIA} \quad [10]$$

Using zero emissions simulations to quantify the inertia term, our novel approach isolates the response to negative emissions by taking the difference between the ramp-down phase of the CDR-reversibility simulation and the zero emissions simulation for a particular mode e.g., the fully coupled mode is shown below:

240

$$\Delta C_L^{NE} = \Delta C_L - \Delta C_L^{INERTIA} = \beta_L(\Delta C_A - \Delta C_A^{INERTIA}) + \gamma_L(\Delta T - \Delta T^{INERTIA}) = \beta_L(\Delta C_A^{NE}) + \gamma_L(\Delta T^{NE}) \quad [11]$$

$$\Delta C_O^{NE} = \Delta C_O - \Delta C_O^{INERTIA} = \beta_O(\Delta C_A - \Delta C_A^{INERTIA}) + \gamma_O(\Delta T - \Delta T^{INERTIA}) = \beta_O(\Delta C_A^{NE}) + \gamma_O(\Delta T^{NE}) \quad [12]$$

As the CDR-reversibility simulation is concentration-driven, carbon gained or lost by the land or ocean does not affect the
245 atmospheric $CO_2$ concentration and surface air temperature as would be expected in the real world. Therefore, we assume that the "true" change in atmospheric $CO_2$ concentration in the ramp-down simulation is the sum of the change in atmospheric $CO_2$ concentration in the "CDR-reversibility" ramp-down phase and [2] the change in the atmospheric $CO_2$ concentration due to the response of the land and ocean, which is further decomposed, into a correction for carbon pools responding to the change in $CO_2$ concentration in the ramp-down phase (rather than to the "true" change in $CO_2$ concentration), and an inertia term. The
250 same is assumed for the surface air temperature:

$$\Delta \mathbb{C}_A = \Delta C_A + \Delta C_A^{(L+O)} = \Delta C_A + \Delta C_A^{(DIFF)} + \Delta C_A^{(INERTIA)} \quad [13]$$

$$\Delta \mathbb{T}_A = \Delta T + \Delta T^{(L+O)} = \Delta T + \Delta T^{(DIFF)} + \Delta T^{(INERTIA)} \quad [14]$$

255 Assuming that [3] $\Delta C_A^{(DIFF)}$ and $\Delta T^{(DIFF)}$ are negligible:

$$\Delta \mathbb{C}_A = \Delta C_A + \Delta C_A^{(INERTIA)} \quad [15]$$

$$\Delta \mathbb{T}_A = \Delta T + \Delta T^{(INERTIA)} \quad [16]$$

We quantify the change in atmospheric $CO_2$ and temperature due to negative emissions alone as difference between the "true"
260 change in change in atmospheric $CO_2$ concentration and the inertia term:

**H** Number: 1  Author:  Subject:Highlight  Date:2023-03-14 8:56:25 AM

When negative emissions are applied

Author:  Subject:Note  Date:2023-03-14 9:57:15 AM

Note that in your case, you are not applying negative emissions but rather $CO_2$ concentration is specified.

**H** Number: 2  Author:  Subject:Highlight  Date:2023-03-14 8:56:37 AM

the change in the atmospheric $CO_2$ concentration due to the
response of the land and ocean,

Author:  Subject:Note  Date:2023-03-14 8:56:44 AM

Are you assuming that the response of land and ocean doesn't change diagnosed emissions?

**H** Number: 3  Author:  Subject:Highlight  Date:2023-03-14 8:56:53 AM

$\Delta CA$ (DIFF)
and $\Delta T$(DIFF)

Author:  Subject:Note  Date:2023-03-14 8:57:01 AM

These two terms haven't been defined? I am not sure what they represent.

[revised manuscript text omitted]
 then decreasing CO2 concentrations" since your simulations are concentration driven and you haven't shown diagnosed emissions.

**Number: 2  Author:  Subject:Highlight  Date:2023-03-14 8:57:25 AM**

emaining in vegetation and 170 PgC remaining in the soil (figure 5(a, c, d)), whereas
the ocean carbon pool holds much more carbon (615PgC) than at preindustrial (figure 5(b)

> **Author:  Subject:Note  Date:2023-03-14 9:58:25 AM**
>
> This is expected given the time lags associated with cVeg and cSoil.

**Number: 3  Author:  Subject:Highlight  Date:2023-03-14 9:50:25 AM**

though this response would not be
405 expected given the asymmetric surface air temperature response in this mode

> **Author:  Subject:Note  Date:2023-03-14 9:50:32 AM**
>
> Even if, temperature response were perfectly symmetric we would still see hysteresis due to time lags associated with the turnover time in cVeg and cSoil pools.

[revised manuscript text omitted]

[Figure]
 Author:  Subject:Note  Date:2023-03-14 9:50:50 AM

This and several other changes described in C fluxes are hard to follow looking at Figure 7 because Figure 7 shows pools and not fluxes.

[Figure]

**Figure 7: a. Atmospheric CO₂ concentration anomaly b. surface air temperature change c. land carbon change d. vegetation carbon change e. soil carbon change and f. ocean carbon change for the zero emissions simulations relative to 1850 (preindustrial). ALL = the CDR-reversibility ramp-up phase from which all modes are initialized; BGC = biogeochemically coupled, RAD = radiatively coupled and FULL = fully coupled. Solid lines are for the ramp-up phase; dashed lines are for the zero emissions phase.**

**3.2.2 "Ramp-down – Zeroemit" Approach: Isolating the Response to Negative Emissions**

The "Ramp-down – Zeroemit" approach uses the zero emissions simulations described in the previous section to isolate the response to negative emissions in the "CDR-reversibility" simulations by taking the difference between the ramp-down phase of the RAD (BGC) "CDR-reversibility" simulation and the RAD (BGC) zero emissions simulation. In the BGC mode, despite our attempt to reduce climate system inertia in our novel approach, carbon pools do not return to their preindustrial states at the time atmospheric CO₂ returns to preindustrial levels **(figure 5)**. In the RAD mode, all carbon pools gain more carbon than they held at preindustrial **(figure 6)**.

The "Ramp-down – Zeroemit" approach removes the initial carbon increase in the "CDR-reversibility" BGC mode **(figure 5)** and removes the initial carbon decrease in the "CDR-reversibility" RAD mode **(figure 6)** reducing the width of the hysteresis. Zickfeld et al. (2016) used zero emissions to isolate the response to negative emissions and observed a reduction in the initial carbon change at the beginning of the ramp-down phase consistent with our results. One possible reason why the hysteresis persists may be irreversible changes in vegetation distribution in the "CDR-reversibility" ramp-down phase that are caused by state changes rather than inertia. When negative emissions are applied, the earth system is in a state of elevated CO₂ concentration and surface air temperature, which may lead to a different vegetation response than to an equivalent amount of

Number: 1 Author: Subject:Highlight Date:2023-03-14 9:50:59 AM
When negative emissions are applied

Author: Subject:Note Date:2023-03-14 9:51:06 AM
Note that these simulations are concentration driven.

[revised manuscript text omitted]

---

## Author Response (AR2)

Response to Anonymous Referee #2

**PAGE 1**

I find this confusing to understand. Note that feedback parameters are a way to show the reponse of the system in a simple way to CO2 and T changes. Given their well known scenario-dependence I am not sure if people use them to find land and ocean C gain back from another scenario. If you are just say that the feeback parameters are different between yours and the standard approach that's sufficient in my view.

Again same thing can go here. Can you highlight examples where people use feedback parameters to calculate emissions?

- *We thank the reviewer for their comment. We have updated the last section of the abstract to make this clearer.*

**PAGE 6**

This has to mention radiation too.

Please consider rewording this. It's better to say - "radiation module sees a specified time-invariant $CO_2$ concentration".

- *We have revised the descriptions for the fully coupled, biogeochemically coupled and radiatively coupled modes accordingly.*

**PAGE 8**

The carbon-cocentration feedback is -ve from the atmosphere's perspective, but positive from the land and ocean's perspective. Please keep this distinction. Here in this sentence you are talking from feedback parameters for land and ocean. That is, beta_A = -(beta_L + beta_O). I see this is done at the end of this paragraph but please mention this earlier.

- *We have clarified the meaning of the feedback parameters earlier in the paragraph.*

This is all very confusing, and the result of expressing change in CO2 concentration related to the peak. If change in CO2 concentration is found relative to PI, then change in CO2 concentration is still +ve until it becomes zero when CO2 gets down to its PI value again. Please consider making this clear that the sign reversal is the result of how you are starting at the peak and not at the PI value.

- *We thank the reviewer for this comment. We have clarified this in the text.*

**PAGE 9**

Note that in your case, you are not applying negative emissions but rather CO2 concentration is specified.

- *We have corrected this throughout the manuscript.*

Are you assuming that the response of land and ocean doesn't change diagnosed emissions?

These two terms haven't been defined? I am not sure what they represent. ($\Delta$CA (DIFF) and $\Delta$T(DIFF))

- *We have reworked the equations for our novel approach (see Section 2.4.1) to make them easier to understand and follow. We no longer include the terms $\Delta CA$ (DIFF) and $\Delta T$(DIFF) in our reasoning.*

**PAGE 15**

I would rather say "is a combination of the response to both increasing and then decreasing CO2 concentrations" since your simulations are concentration driven and you haven't shown diagnosed emissions.

- *Done.*

This is expected given the time lags associated with cVeg and cSoil

Even if, temperature response were perfectly symmetric we would still see hysteresis due to time lags associated with the turnover time in cVeg and cSoil pools.

- *We thank the reviewer for this comment. We have corrected this in the text.*

**PAGE 18**

This and several other changes described in C fluxes are hard to follow looking at Figure 7 because Figure 7 shows pools and not fluxes.

- *We have revised most of the manuscript to describe carbon pool changes instead of carbon flux changes so that the text is consistent with the figures shown.*

**PAGE 19**

Note that these simulations are concentration driven.

- *We have corrected this.*

**PAGE 21**

 replaced with "simulations"

- *Done.*

**PAGE 22**

I think, you mean mineralization.

- *Done.*

[inserted text: to] due ^ changes in climate alone

- *Done.*